# Can Push-forward Generative Models Fit Multimodal Distributions?

**Antoine Salmona**
Centre Borelli,
ENS Paris Saclay, France

**Agnès Desolneux**
Centre Borelli, CNRS
ENS Paris Saclay, France

**Julie Delon**
MAP5, Université Paris Cité, France
Institut Universitaire de France (IUF)

**Valentin De Bortoli**
Center for Sciences of Data, CNRS
ENS Ulm, France

## Abstract

Many generative models synthesize data by transforming a standard Gaussian random variable using a deterministic neural network. Among these models are the Variational Autoencoders and the Generative Adversarial Networks. In this work, we call them "push-forward" models and study their expressivity. We formally demonstrate that the Lipschitz constant of these generative networks has to be large in order to fit multimodal distributions. More precisely, we show that the total variation distance and the Kullback-Leibler divergence between the generated and the data distribution are bounded from below by a constant depending on the mode separation and the Lipschitz constant. Since constraining the Lipschitz constants of neural networks is a common way to stabilize generative models, there is a provable trade-off between the ability of push-forward models to approximate multimodal distributions and the stability of their training. We validate our findings on one-dimensional and image datasets and empirically show that the recently introduced diffusion models do not suffer of such limitation.

## 1 Introduction

Generative modeling has become over the last years one of the most popular research topics in machine learning and computer vision. From a mathematical perspective, the goal of generative modeling can be seen as predicting new synthetic samples from an unknown probability distribution $\nu$ on $\mathbb{R}^d$ given the information of $m$ *true* samples $x_i$ drawn from $\nu$ (the data distribution). A general approach to solve this problem is to define a parametric family of probability distributions $(\nu_\theta)_{\theta \in \Theta}$ and solve the problem

$$\min_{\theta \in \Theta} D(\tfrac{1}{m} \sum_i \delta_{x_i}, \nu_\theta) \,,$$

where $D$ is a similarity measure between probability distributions and $\delta_x$ is the delta distribution at $x$. Beside their direct application (Sandfort et al., 2019; Antoniou et al., 2018), generative models have been used in numerous applications in various machine learning subfields, such as solving inverse problems (Ravuri et al., 2021; Ledig et al., 2017) or machine translation (Isola et al., 2017; Yang et al., 2018). However, most generative modeling methods still lack theoretical understanding and it remains often unclear whether the method approaches correctly the probability distribution $\nu$ or only generates samples that appear to have been drawn from $\nu$ without fully recovering the underlying structure of the distribution. In this work, we focus on the particular class of *push-forward generative models*. Those models have in common that for any $\theta \in \Theta$, the parametric distribution $\nu_\theta$ approaching $\nu$ is of the form

$$\nu_\theta = g_{\theta \#} \mu_p \,,$$

where $\mu_p = \mathrm{N}(0, \mathrm{Id}_p)$ is the Gaussian standard distribution in dimension $p$, $\#$ is the push-forward operator [1], and $g_\theta : \mathbb{R}^p \to \mathbb{R}^d$ is a deterministic neural network of parameter $\theta$. This class includes two of the most popular generative models: the *Variational Auto-Encoders* (VAEs) [2] (Kingma and Welling, 2014) and the *Generative Adversarial Networks* (GANs) (Goodfellow et al., 2014). It also includes other models such as most of normalizing flows (Rezende and Mohamed, 2015).

Deep neural networks are most of the time Lipschitz mappings by design, since their activation functions are generally Lipschitz. In the literature, constraining the Lipschitz constant of a neural network is widely used as a way to increase its robustness (Virmaux and Scaman, 2018; Fazlyab et al., 2019), in particular to adversarial attacks (Goodfellow et al., 2015). Common approaches to bound Lipschitz constants of neural networks are spectral normalization (Miyato et al., 2018), adding a gradient penalization in the loss (Gulrajani et al., 2017; Mohajerin Esfahani and Kuhn, 2018), or Jacobian regularization (Pennington et al., 2017). These approaches have been widely used to stabilize the training of GANs, where Lipschitz constraints have been first imposed on discrimators (Arjovsky et al., 2017; Kodali et al., 2017; Fedus et al., 2018), while recent state-of-the-art architectures such as BigGAN (Brock et al., 2018), SAGAN (Zhang et al., 2019) or StyleGAN2 (Karras et al., 2020) also impose similar constraints on the generators through spectral normalization (Brock et al., 2018; Zhang et al., 2019), or Jacobian regularization (Karras et al., 2020). In contrast to GANs, the recent study of Kumar and Poole (2020) shows that the decoder Jacobian in VAEs is implicitly regularized, which limits its Lipschitz constant.

Recently, Dhariwal and Nichol (2021) trained an unconditional *Score-based Generative Model* (SGM) (Song and Ermon, 2019; Ho et al., 2020) on ImageNet (Russakovsky et al., 2015) and achieved state-of-the-art generation. To the best of our knowledge, there is no push-forward generative model capable of reaching this kind of performance on such a complex dataset without explicitly adding any conditional label information in the model, see (Brock et al., 2018) for instance. SGMs (also known as *diffusion models*) proceed as follows: first, noise is progressively added to the data distribution until we reach a standard Gaussian distribution. Then this forward dynamics is reversed leveraging recent advances in deep learning and tools from score-matching (Hyvärinen, 2005; Vincent, 2011). We refer to Song et al. (2020) for an introduction on SGMs. In those models, the parametric distribution is also of the form $\nu_\theta = g_{\theta\#}\mu_p$, where $g_\theta$ is the whole reverse diffusion dynamic (which can be seen as a composition of deterministic Lipschitz mappings) and $p = d(N + 1)$ with $N$ being the total number of steps in the dynamic (More details can be found in Appendix S2). However, those models are not push-forward generative models in the strict sense of the term, since the push-forward mapping is not a simple neural network anymore. An important difference is that optimization is not directly performed on the push-forward mapping itself but on an auxiliary function (the score). We therefore categorize them as *indirect push-forward generative models* in this work.

**Contributions of the paper.** In this paper, we study the expressivity of direct and indirect push-forward generative models in relation to the Lipschitz constant of the push-forward mapping they learn. More precisely, in Section 3, for a Lipschitz function $g$ and a given multimodal probability distribution $\nu$, we formally demonstrate that the Lipschitz constant of $g$ must necessarily be large in order for $g_\#\mu_p$ to approximate $\nu$ correctly, as it has been already intuitively observed in the literature (Lu et al., 2020; Luise et al., 2020; Khayatkhoei et al., 2018). As a direct consequence, we exhibit lower bounds on $D(g_\#\mu_p, \nu)$, where $D$ is the total variation distance or the Kullback-Leibler divergence, with an explicit dependence on the Lipschitz constant $\mathrm{Lip}(g)$ of $g$, which highlights that there is a fundamental trade-off for (direct) push-forward generative models between expressivity and stability of training. In Section 4, we illustrate these theoretical results on several experiments, showing the difficulties of GANs and VAEs to simulate multimodal distributions. We compare these models with SGMs and show experimentally that SGMs seem to be able to generate correctly multimodal distributions while keeping the Lipschitz constant of the score network relatively small, suggesting that these models do not suffer of such previously mentioned limitations. All the proofs are postponed to the appendix.

---

[1] If $\mu$ is a measure on $\mathbb{R}^p$ and $f$ is a mapping from $\mathbb{R}^p$ to $\mathbb{R}^d$, the push-forward measure $f_\#\mu$ is the measure on $\mathbb{R}^d$ such that for all Borel set $\mathsf{A}$ of $\mathbb{R}^d$, $f_\#\mu(\mathsf{A}) = \mu(f^{-1}(\mathsf{A}))$.

[2] In this work, the VAE model considered is the Gaussian-VAE since the data are real-valued. See Appendix S2 for details on why the generated distribution is of the form $g_{\theta\#}\mu_p$ in this model.

## 2 Related Works

Assessing the efficiency of push-forward models is a recurrent and important question in the literature. Sajjadi et al. (2018) and Kynkäänniemi et al. (2019) propose Precison and Recall metrics to assess GANs, aiming to measure simultaneously the mode collapse and the proportion of off-manifold generated samples. Using similar metrics, Tanielian et al. (2020) prove an upper bound on the precision of vanilla GANs (the proportion of generated samples which could have been generated by the target distribution). To overcome this limitation, they simply propose to reject samples associated with large values of the generator Jacobian. The intuition behind this idea is that those samples lie in regions of the space where the discontinuous optimal generator would "jump" between modes and so are off-manifold.

In the context of normalizing flows, it has been shown that the invertibility constraint limits the expressivity of the model. Indeed, Cornish et al. (2020) show that distributions generated by invertible normalizing flows have a support which is necessarily homeomorphic to the support of the latent distribution. As an outcome, the Lipschitz constant of the inverse flow has to approach infinity to correctly approximate distributions lying on disconnected manifolds (Cornish et al., 2020; Hagemann and Neumayer, 2021; Behrmann et al., 2021). To improve the expressivity of normalizing flows, it has been proposed in Cornish et al. (2020) and Wu et al. (2020) to inject stochasticity in the model.

Another line of research focuses on the fact that the model has access to only the empirical distribution $\nu_n = \frac{1}{n} \sum_i \delta_{x_i}$ and not to the true target distribution. For instance Nagarajan et al. (2018) study to what extend GANs only memorize the data. Gulrajani et al. (2018) highlight the fact that common GAN benchmarks prefer training set memorization to a model which imperfectly fits the true distribution but covers more of its support. Related to this, Stéphanovitch et al. (2022) study specifically the Wasserstein GAN case, where the latent distribution is uniform and construct an optimal generator which minimizes the Wasserstein distance of order 1 between the push-forward measure and the empirical distribution, thus deriving a lower bound on the 1-Wasserstein distance. In the same paper, and more related to our work, the authors study the asymptotic case of an infinite number of data and show that most of the time the minimal 1-Wasserstein distance between the push-forward measure and the target distribution remains strictly positive.

## 3 Push-forward measure and Lipschitz mappings

In this section, we study the properties of the push-forward measure $g_{\#}\mu_p$ when $\mu_p = \mathrm{N}(0, \mathrm{Id}_p)$ is the standard Gaussian distribution in dimension $p$ and $g$ is a Lipschitz mapping. First, for any probability measure $\gamma$ on $\mathbb{R}^d$ and any Borel set $\mathsf{A}$ of $\mathbb{R}^d$, we define the $\gamma$-surface area of $\mathsf{A}$ by

$$\gamma^+(\partial \mathsf{A}) = \liminf_{\varepsilon \to 0^+} (\gamma(\mathsf{A}_\varepsilon) - \gamma(\mathsf{A}))/\varepsilon \ ,$$

where $\mathsf{A}_\varepsilon = \{x \in \mathbb{R}^d \ : \ \text{there exists } a \in \mathsf{A}, \ \|x - a\| \le \varepsilon\}$ is the $\varepsilon$-extension of $\mathsf{A}$ and $\partial \mathsf{A}$ is the boundary of $A$. The $\gamma$ - surface area can be interpreted as the mass of $\gamma$ on the hypersurface $\partial \mathsf{A}$. Note that the support of $\gamma$ and $A$ can be sets of intrinsic dimension smaller than $d$, which is most of the time the case when working with real data which are likely to live on low dimensional manifolds (Pope et al., 2020). The main theoretical result of this paper establishes some properties of push-forward measures depending on the regularity of the push-forward mapping.

**Theorem 1.** *Let $g : \mathbb{R}^p \to \mathbb{R}^d$ be a Lipschitz function with Lipschitz constant $\mathrm{Lip}(g)$. Then for any Borel set $\mathsf{A} \in \mathcal{B}(\mathbb{R}^d)$,*

$$\mathrm{Lip}(g)(g_{\#}\mu_p)^+(\partial \mathsf{A}) \ge \varphi\left(\Phi^{-1}(g_{\#}\mu_p(\mathsf{A}))\right) \ , \tag{1}$$

*where $\varphi(x) = (2\pi)^{-1/2} \exp[-x^2/2]$ and $\Phi(x) = \int_{-\infty}^x \varphi(t)\mathrm{d}t$. In addition, we have that for any $r \ge 0$*

$$g_{\#}\mu_p(\mathsf{A}_r) \ge \Phi\left(r/\mathrm{Lip}(g) + \Phi^{-1}(g_{\#}\mu_p(\mathsf{A}))\right) \ . \tag{2}$$

*Sketch of proof.* The proof of this result consists in establishing lower-bounds on $(g_{\#}\mu_p)^+(\partial \mathsf{A})$ and $g_{\#}\mu_p(\mathsf{A}_r)$ which can be expressed as Gaussian integrals. We conclude upon combining this result and the Gaussian isoperimetric inequality, see Sudakov and Tsirelson (1978) $\mu_p^+(\partial \mathsf{A}) \ge \varphi(\Phi^{-1}(\mu_p(\mathsf{A})))$. $\qquad \square$

Note that (2) implies (1) upon remarking that (2) is an equality for $r = 0$, dividing by $r$ and letting $r \to 0$. Theorem 1 recovers the Gaussian inequality in the case where $g$ is the identity mapping and extends it to all Lipschitz mappings. As the Gaussian inequality, Theorem 1 is dimension free, in the sense that neither $d$, nor $p$, nor the intrinsic dimension of $g(\mathbb{R}^p)$ play a role in the lower bounds. In the following section, we are going to use Theorem 1 to (i) give a lower bound on the Lipschitz constant so that push-forward generative models *exactly* match the data distribution, (ii) give a lower bound on the total variation and the Kullback-Leibler divergence between the push-forward and data distributions which depends on the Lipschitz constant of the model.

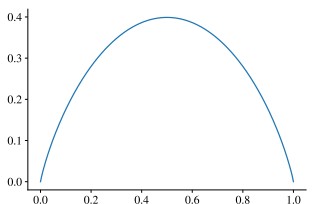

Figure 1: Graph of $\varphi \circ \Phi^{-1}$.

### 3.1 Lower bounding the Lipschitz constant of push-forward mappings

Equation (1) implies that the Lipschitz constant of $g$ must necessarily be large for $g_{\#}\mu_p$ to be multimodal. It provides indeed a lower bound on the Lipschitz constant of the mappings $g$ which push $\mu_p$ into a given measure $\nu$. In the extreme case where the support of $\nu$ is composed of disconnected manifolds, we retrieve that there doesn't exist any Lipschitz mapping which pushes $\mu_p$ into $\nu$ since it can be found Borel sets A with null $\nu$-surface area but such that the right-hand term of (1) is strictly positive (which occurs when $0 < \nu(A) < 1$). In the intermediate case where the support of $\nu$ is connected but $\nu$ is multimodal, the less mass $\nu$ has between modes, the larger must be the Lipschitz constant of the mappings which push $\mu_p$ into $\nu$. Indeed, if $\nu$ has little mass between its modes, one can find sets A with small $\nu$-surface area and such that $0 < \nu(A) < 1$. As a toy example, we get an explicit bound on the Lipschitz constant of the mappings which push $\mu_p$ into a mixture of two isotropic Gaussians.

**Corollary 2.** *Let $\nu = \lambda \mathrm{N}(m_1, \sigma^2 \mathrm{Id}_d) + (1 - \lambda)\mathrm{N}(m_2, \sigma^2 \mathrm{Id}_d)$ with $m_1, m_2 \in \mathbb{R}^d$, $\sigma > 0$ and $\lambda \in (0, 1)$. Assume that there exists $g : \mathbb{R}^p \to \mathbb{R}^d$ Lipschitz such that $g_{\#}\mu_p = \nu$. Then*

$$\mathrm{Lip}(g) \geq \sigma \exp \left[ \|m_2 - m_1\|^2/(8\sigma^2) - (\Phi^{-1}(\lambda))^2/2 \right] .$$

*Sketch of proof.* The proof of this result consists into applying Inequality (1) of Theorem 1 on the half-space H such that $\partial$H is the equidistant line of the means of the two Gaussians, and then to explicit the value of $\nu^+(\partial$H$)$. See Figure S1 for a visualization of $\partial$H in the univariate case. □

Note that assuming there exists $g : \mathbb{R}^p \to \mathbb{R}^d$ such that $g_{\#}\mu_p = \nu$ implies $p \geq d$ since $\nu$ covers the whole ambient space and so $g$ must be a surjective mapping. This bound is maximal in the balanced case when $\lambda = 1/2$ since $\Phi^{-1}(\lambda) = 0$ in that case. Otherwise, the more unbalanced the modes are, the smaller the bound is since the two terms in the exponential compensate each other more and more. Extending this corollary to mixtures of more than two Gaussians with different covariance matrices is technically difficult but we could expect a similar exponential growth in the square distance between modes since it depends mainly on the order of magnitude of the local minima of the distribution density. As a by product of Theorem 1, we also get the following result which shows that (in the one-dimensional case) the optimal transport map for the $\ell_2$ cost minimizes the Lipschitz constant of the push-forward mapping.

**Corollary 3.** *Let $\nu$ be a probability measure on $\mathbb{R}$ with density w.r.t. the Lesbesgue measure and such that $\mathrm{supp}(\nu) = \mathbb{R}$. Assume that there exists $g : \mathbb{R}^p \to \mathbb{R}$ Lipschitz such that $\nu = g_{\#}\mu_p$. Let us denote $T_{\mathrm{OT}} = \Phi_{\nu}^{-1} \circ \Phi$ the Monge map between $\mu_1$ and $\nu$, where $\Phi_{\nu}$ is the cumulative distribution function of $\nu$. Then we have $\mathrm{Lip}(g) \geq \mathrm{Lip}(T_{\mathrm{OT}})$.*

To the best of our knowledge, extending this proposition to the case where $d > 1$ remains an open problem. We show now that Equation (2) of Theorem 1 allows to derive lower bounds on similarity measures between the push-forward measure and the target distribution.

### 3.2 Lower bounds on similarity measures between probability distributions

Equation (2) provides a bound on the minimal mass the push-forward measure $g_{\#}\mu_p$ can have on a given set when $g$ is fixed with Lipschitz constant $\mathrm{Lip}(g)$. As a consequence, if $\nu$ is a distribution

such that there exists sets on which $\nu$ has less mass than the minimal quantity that $g_{\#}\mu_p$ can reach on those sets given the value of $\mathrm{Lip}(g)$, then $g_{\#}\mu_p$ cannot be equal to $\nu$, implying that most of similarity measures between $g_{\#}\mu_p$ and $\nu$ will be automatically strictly positive. In the following, we consider that $g$ and $\nu$ are fixed and we derive lower bounds on the total variation distance and the Kullback-Leibler divergence between $g_{\#}\mu_p$ and $\nu$. We recall that the total variation distance between two probability measures on $\mathbb{R}^d$, $\nu_0, \nu_1$ is given by

$$d_{\mathrm{TV}}(\nu_0, \nu_1) = \sup\{\nu_0(\mathsf{A}) - \nu_1(\mathsf{A}) \ : \ \mathsf{A} \in \mathcal{B}(\mathbb{R}^d)\} \ .$$

Similarly, we define the Kullback-Leibler divergence between two probability measures on $\mathbb{R}^d$, $\nu_0, \nu_1$, using the Donsker-Varadhan representation (Dupuis and Ellis, 2011, Lemma 1.4.3a):

$$d_{\mathrm{KL}}(\nu_0 || \nu_1) = \sup\{\textstyle\int_{\mathbb{R}^d} f(x)\mathrm{d}\nu_0(x) - \log\left(\int_{\mathbb{R}^d} \exp[f(x)]\mathrm{d}\nu_1(x)\right) \ : \ f \in \mathfrak{B}(\mathbb{R}^d, \mathbb{R})\} \ ,$$

where $\mathfrak{B}(\mathbb{R}^d, \mathbb{R})$ denotes the set of all bounded mappings from $\mathbb{R}^d$ to $\mathbb{R}$. In the following, we will denote for any $\mathsf{A} \in \mathcal{B}(\mathbb{R}^d)$ and $r > 0$,

$$\alpha_g(\mathsf{A}, r) = \Phi\left(r/\mathrm{Lip}(g) + \Phi^{-1}(g_{\#}\mu_p(\mathsf{A}))\right) \ ,$$
$$\beta_g(\mathsf{A}, r) = \alpha_g(\mathsf{A}, r) - g_{\#}\mu_p(\mathsf{A}) \ ,$$

where $\alpha_g(\mathsf{A}, r)$ and $\beta_g(\mathsf{A}, r)$ are the lower bounds of $g_{\#}\mu_p(\mathsf{A}_r)$ and $g_{\#}\mu_p(\mathsf{A}_r \setminus \mathsf{A})$ provided by Theorem 1. We start by proving lower bounds on the total variation distance.

**Theorem 4.** *Let $\nu$ be a probability measure on $\mathbb{R}^d$ and let $g : \mathbb{R}^p \to \mathbb{R}^d$ be a Lipschitz function. Then,*

$$d_{\mathrm{TV}}(g_{\#}\mu_p, \nu) \geq \sup\{\alpha_g(\mathsf{A}, r) - \min\{g_{\#}\mu_p(\mathsf{A}), \nu(\mathsf{A})\} - \nu(\mathsf{A}_r \setminus \mathsf{A}) \ : \ \mathsf{A} \in \mathcal{B}(\mathbb{R}^d), r > 0\} \ . \quad (3)$$

*Sketch of Proof.* The proof of this result consists in bounding from below the total variation distance on one hand by $|g_{\#}\mu_p(\mathsf{A}_r \setminus \mathsf{A}) - \nu(\mathsf{A}_r \setminus \mathsf{A})|$ and on the other hand by $|g_{\#}\mu_p(\mathsf{A}_r) - \nu(\mathsf{A}_r)|$ for a given $\mathsf{A} \in \mathcal{B}(\mathbb{R}^d)$ and a given $r > 0$, and then applying Theorem 1. $\square$

Observe that (3) always holds but the right-hand term may become negative if the Lipschitz constant of $g$ is large enough. The main idea behind this bound is to find a set $\mathsf{A}$ and a real $r > 0$ such that $\nu$ has a lot of mass on $\mathsf{A}$ but almost no mass on $\mathsf{A}_r \setminus \mathsf{A}$. For instance, if $\nu$ is a distribution on two disconnected manifolds $\mathsf{M}_1$ and $\mathsf{M}_2$, an optimal choice for $\mathsf{A}$ would either be $\mathsf{M}_1$ or $\mathsf{M}_2$ and the optimal $r$ would be the distance between the two manifolds. Using Theorem 4, one can derive smaller but more explicit lower bounds only depending on $\nu$ and the Lipschitz constant of $g$. As a first example, we derive an explicit lower bound in the case where $\nu$ is a bi-modal distribution on two disconnected manifolds.

**Corollary 5.** *Let $\nu$ be a measure on $\mathbb{R}^d$ on two disconnected manifolds $\mathsf{M}_1$ and $\mathsf{M}_2$ such that $\nu(\mathsf{M}_1) = \lambda$ and $\nu(\mathsf{M}_2) = 1 - \lambda$, with $\lambda \in [1/2, 1)$, and let $g : \mathbb{R}^p \to \mathbb{R}^d$ be a Lipschitz function. Then,*

$$d_{\mathrm{TV}}(g_{\#}\mu_p, \nu) \geq \textstyle\int_{\Phi^{-1}(\lambda)}^{d(\mathsf{M}_1, \mathsf{M}_2)/2\mathrm{Lip}(g) + \Phi^{-1}(\lambda)} \varphi(t)dt \ ,$$

*where $d(\mathsf{M}_1, \mathsf{M}_2) = \inf\{\|m_1 - m_2\| \ : \ m_1 \in \mathsf{M}_1, m_2 \in \mathsf{M}_2\}$.*

As a second example, we also get an explicit lower bound in the case where $\nu$ is a mixture of two isotropic Gaussians. For simplicity we stick to the balanced case.

**Corollary 6.** *Let $\nu = (1/2)[\mathrm{N}(m_1, \sigma^2 \mathrm{Id}_d) + \mathrm{N}(m_2, \sigma^2 \mathrm{Id}_d)]$ with $m_1, m_2 \in \mathbb{R}^d$ and $\sigma \geq 0$. Let $g : \mathbb{R}^p \to \mathbb{R}^d$ be a Lipschitz function. Then,*

$$d_{\mathrm{TV}}(g_{\#}\mu_p, \nu) \geq \textstyle\int_0^{\|m_2 - m_1\|/4\sigma\mathrm{Lip}(g)} \varphi(t)dt - (1/2)\int_{\|m_2 - m_1\|(2\sigma-1)/4\sigma^2}^{\|m_2 - m_1\|(2\sigma+1)/4\sigma^2} \varphi(t)dt \ .$$

In both corollaries, the lower bound tends to $1/2$ when the distance between the modes tends to infinity, meaning that $g_{\#}\mu_p$ is far from well approaching $\nu$. Note that the lower bound exhibited in Corollary 5 is always strictly positive regardless of the value of the Lipschitz constant of $g$. One can also observe that this latter bound is maximal in the balanced case, when $\lambda = 1/2$, since the standard normal distribution concentrates its mass around $0$. Finally, we end this section by deriving a similar lower bound on the Kullback-Leibler divergence between $g_{\#}\mu_p$ and $\nu$. We consider the Kullback-Leibler divergence since this is a measure of similarity between measures which is bounded and is very sensitive to the mismatch of supports between the generated and the data distributions.

**Theorem 7.** *Let $\nu$ be a probability measure on $\mathbb{R}^d$ and let $g : \mathbb{R}^p \to \mathbb{R}^d$ be a Lipschitz function. Then,*

$$d_{\mathrm{KL}}(g_{\#}\mu_p||\nu) \geq \sup\{\beta_g(\mathsf{A},r)\log\left(\tfrac{\beta_g(\mathsf{A},r)}{\nu(\mathsf{A}_r\setminus\mathsf{A})}\right) + (1-\beta_g(\mathsf{A},r))\log\left(\tfrac{1-\beta_g(\mathsf{A},r)}{1-\nu(\mathsf{A}_r\setminus\mathsf{A})}\right) \, : \, \mathsf{A}\in\mathcal{B}(\mathbb{R}^d), r>0\} \, .$$

*Sketch of Proof.* The proof of this result consists in setting $f = \zeta\chi_{\mathsf{A}_r\setminus\mathsf{A}}$ with $\zeta > 0$ and where $\chi_{\mathsf{A}}$ is the characteristic function of the set $\mathsf{A}$ and plugging it in the Donsker-Varadhan representation in order to get a lower bound depending on the probability $g_{\#}\mu_p(\mathsf{A}_r \setminus \mathsf{A})$ for a given $\mathsf{A}$, a given $r$ and a given $\zeta$. Then we apply Theorem 1 and we derive the optimal value of $\zeta$. □

As above, this bound always holds but the right-hand term becomes negative if $\mathrm{Lip}(g)$ is large enough. As for Theorem 4, the main idea is to find a set $\mathsf{A}$ and a real $r$ such that $\nu$ has a lot of mass on $\mathsf{A}$, but $\nu$ has almost no mass on $\mathsf{A}_r \setminus \mathsf{A}$. Observe that if $\nu(\mathsf{A}_r \setminus \mathsf{A})$ tends to 0, the left-hand term of the bound tends to infinity. This is coherent with the behavior of the Kullback-Leibler divergence. Similarly to Corollary 6, we also get an explicit lower bound in the case where $\nu$ is a mixture of two isotropic Gaussians. As for Corollary 6, we stick to the balanced case for simplicity.

**Corollary 8.** *Let $\nu = (1/2)\left[\mathrm{N}(m_1, \sigma^2\,\mathrm{Id}_d) + \mathrm{N}(m_2, \sigma^2\,\mathrm{Id}_d)\right]$ with $m_1, m_2 \in \mathbb{R}^d$ and $\sigma \geq 0$. Let $g : \mathbb{R}^p \to \mathbb{R}^d$ be a Lipschitz function. We denote*

$$\lambda = g_{\#}\mu_p\left(\{(m_2 - m_1)^T\left(x - (m_2 + m_1)/2\right) \leq 0 \, : \, x \in \mathbb{R}^d\}\right) \, ,$$

*and we suppose without loss of generality, that $\lambda \in (0, 1/2]$. Then,*

$$d_{\mathrm{KL}}(g_{\#}\mu_p, \nu) \geq A\log\left(\tfrac{A}{B}\right) + (1-A)\log\left(\tfrac{1-A}{1-B}\right) \, ,$$

*where*

$$A = \int_{-\Phi^{-1}(1-\lambda)}^{\|m_2-m_1\|/4\sigma\mathrm{Lip}(g)-\Phi^{-1}(1-\lambda)} \varphi(t)\mathrm{d}t \, , \text{ and } B = (1/2)\int_{\|m_2-m_1\|(2\sigma-1)/4\sigma^2}^{\|m_2-m_1\|(2\sigma+1)/4\sigma^2} \varphi(t)\mathrm{d}t \, .$$

Observe that this time, $\mathrm{Lip}(g)$ is no longer the only dependency in $g$ since the bound also depends on the proportion of the modes of $g_{\#}\mu_p$. However, it should be noted that when $g_{\#}\mu_p$ approximates correctly $\nu$, $\lambda$ is automatically close to $1/2$ and so $\Phi^{-1}(1 - \lambda)$ is small in that case. To conclude, this section, we highlight the fact that, if our results are dimension free in theory, the dimension might be hidden in the distances between modes and the Lipschitz constant of $g$ when working with real datasets. Indeed, the order of magnitude of the Euclidean distance between two samples $x_i$ is likely to increases with the dimension $d$. As an outcome, the orders of magnitude of the distance between modes and so the Lipschitz constant that $g$ must reach for correct generation probably increase with $d$ also.

## 4   Experiments

In what follows, we illustrate the pratical implications of our results by training GANs, VAEs and SGMs on simple bi-modal distributions. More precisely, we show on one hand that generating multimodal distributions with GANs and VAEs is difficult since for those models, good generation necessarily involves generative networks with large Lipschitz constants. On the other hand, we show that SGMs seem to be able to generate multimodal distributions while keeping the Lipschitz constant of the score network relatively small and thus do not suffer of the same limitation. First, we focus on the univariate case where we can easily assess the Lipschitz constants of the networks. Then we illustrate our results in higher dimensions by training the three models on datasets derived from MNIST (LeCun et al., 1998). In all our experiments, we use the same architecture for the VAE decoder and the GAN generator in order to offer rigorous comparisons of the different models. For score-based modeling, we use architectures with similar numbers of learnable parameters. All details on the experiments and architecture of the networks can be found in Appendix S5.

### 4.1   Univariate case

First, we train a VAE and a GAN with one-dimensional latent spaces on 50000 independent samples drawn from a balanced mixture of two univariate Gaussians $\nu = (1/2)[\mathrm{N}(-m, 1) + \mathrm{N}(m, 1)]$ for different values of $m > 0$. We also train a SGM on the same samples.

**Histograms of generated distributions.** Figure 2 shows histograms of generated distributions for $m = 10$ with the three different models. VAE models seem to generate Gaussians modes but interpolate significantly between them, while GANs do not interpolate but fail to retrieve the structure of the target distribution and forget parts of their support, which is known as *mode collapse* and is a common pitfall of such models (Arjovsky and Bottou, 2017; Metz et al., 2017). On the same task, SGMs do not suffer from such shortcomings. In the following section, we will link the interpolation/mode-collapsing properties of these models with their Lipschitz constants.

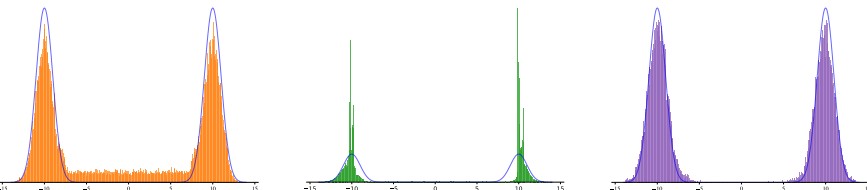

Figure 2: Histograms of distributions generated with VAE (left, in orange), GAN (middle, in green), and with SGM (right, in purple) for $m = 10$. The data distribution densities are plotted in blue.

**Lipschitz constant and mass between modes.** In Figure 3 (right), we observe that the GAN generator reaches much larger Lipschitz constants than the VAE decoder. This explains the difference of behaviors between GAN and VAE observed in Figure 2, as the mapping learned by the VAE is not stiff enough to concentrate the push forward measure on the two modes. One possible explanation for the interpolating behavior of the VAE is that the Euclidean norm of the Jacobian of the VAE decoder is implicitly regularized during training, as it has been demonstrated in Kumar and Poole (2020). Both GAN and VAE saturate the constraint on $g_{\#}\mu_p([-m/2, m/2])$ provided by Theorem 1, meaning that the generative networks minimize the amount of mass between modes as much as their Lipschitz constants allow it. Finally, we can observe that the score network is able to keep a relatively small Lipschitz constant compared to the GAN, while managing to interpolate less than the latter. A probable explanation for this follows from the fact that the score network is used multiple time during inference. Hence, the Lipschitz constant of the push-forward mapping (the whole generation dynamic) is likely much larger than the Lipschitz constant of the neural network itself, and so the model is able to push-forward a Gaussian distribution into a multimodal distribution keeping a relatively small Lipschitz constant of the score network. Finally, in Figure 3 (left), we observe that when $m$ increases, the Lipschitz constant of the VAE decoder and the GAN generator becomes rapidly much smaller than the value of the lower bound provided by Corollary 2. This means that for $m$ large enough it is not possible to close the gap between the data distribution and the push-forward distribution. We highlight that this observation does not apply to SGMs since in this setting the network is applied multiple times.

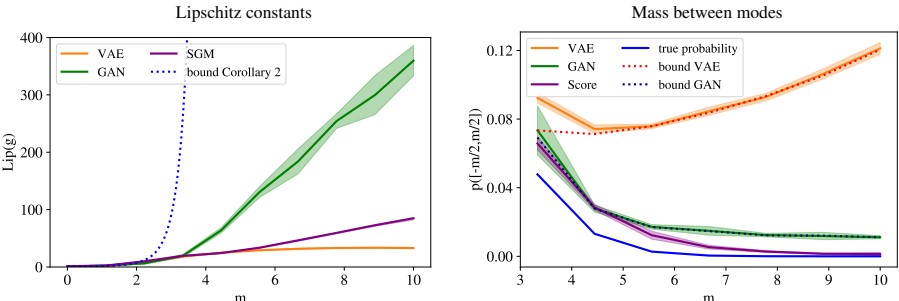

Figure 3: Left: evolution of the Lipschitz constants of the three different generative models trained on 50000 samples of $(1/2)[\mathrm{N}(-m, 1) + \mathrm{N}(m, 1)]$, in function of $m$. Right: evolution of the proportion of samples generated by the three models on the interval $[-m/2, m/2]$. We also show on this graph the lower bounds predicted by Theorem 1 for the VAE and the GAN, as well as the true probability $\nu([-m/2, m/2])$. Experiments are averaged over 10 runs and the colored bands correspond to +/- the standard deviation.

**Stability of GAN and mode collapse.** Odena et al. (2018) suggested that the magnitude of the norm of the generator jacobian may be causally related to instability and mode collapse. This is why many state-of-the-art GANs apply spectral normalization (Miyato et al., 2018) on their generators. In Figure 4 (left), we show that this technique cannot be used when training GANs on multimodal distributions: since spectral normalization constraints the Lipschitz constant of the generator to be smaller than 1, the GAN is trained towards concentrating in one of the modes of the distribution over interpolating massively between them. This has been referred to as *mode dropping* by Khayatkhoei et al. (2018). To complete this analysis, we also train the GAN adding an additional gradient penalty term $10/L^2 \max_{z \sim \mathrm{N}(0, \mathrm{Id}_p)}(\|\nabla_z g_\theta(z)\|_2^2 - L)^2$, in the generator loss function, similarly to WP-GAN (Gulrajani et al., 2017), where $L$ is an hyperparameter corresponding to the targeted Lipschitz constant. As expected, we can observe in Figure 4 (right), that when $\mathrm{Lip}(g)$ increases, the GAN begin to generate both modes but becomes also more and more prone to mode collapse. This illustrates the fundamental trade-off between expressivity and robustness in push-forward generative models.

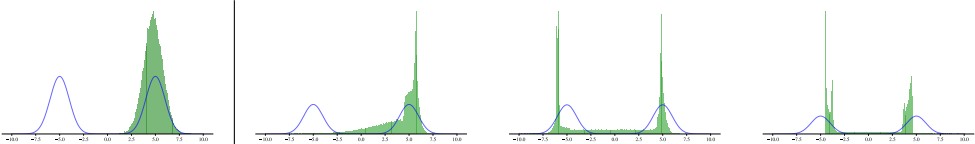

Figure 4: Histograms of distributions generated with GANs with spectral normalization applied on the generator (left), and with gradient penalty (right) for $\mathrm{Lip}(g) \approx L = 5$, $\mathrm{Lip}(g) \approx L = 15$ and $\mathrm{Lip}(g) \approx L = 25$. The data distribution densities are plotted in blue.

**Influences of generator depth and time of training.** In Figure 5, we study the effect of increasing the number of layers of the generative network as well as increasing the training time on the value of the Lipschitz constant of the VAE decoder and the GAN generator. In the VAE setting, the Lipschitz constant increases linearly with the depth of the decoder. This is not the case in the GAN setting, where increasing the size of the model seems to dramatically affect its stability. For both models, the Lipschitz constants of the generative network grow with the number of epochs. Yet this growth seems to be logarithmic for the VAE and the GAN seems to becomes more unstable as the number of epochs increases.

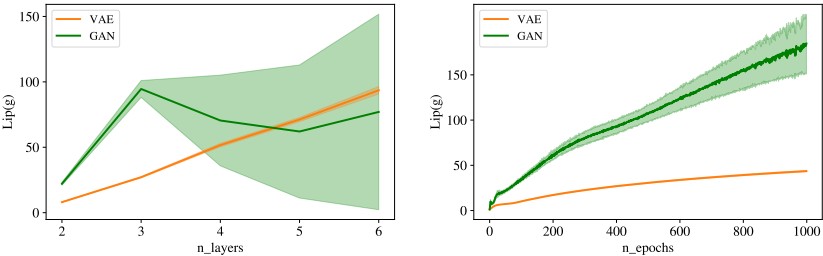

Figure 5: Evolution of the Lipschitz constant of the generative network with respect to its number of layers (left) and of the Lipschitz constant in function of the numbers of epochs (right). The experiments are averaged over 10 runs and the colored bands correspond to +/- the standard deviation.

**Influence of generator architecture.** Finally, we study in Figure 6 the impact of the architecture of the generative network (i.e. the VAE decoder and the GAN generator) on its Lipschitz constant as well as on the training stability of the model by comparing three different architectures: first, we use a simple feed-forward network as precedently, then we add additive skip-connections of type "resnet" (He et al., 2016) to the previous backbone, and last we add concatenation skip-connections of type "densenet" (Huang et al., 2017) instead of additive skip-connections. For both models, it seems that more expressive decoder architectures do not help to reach larger values of Lipschitz constant. However, one can observe that in the GAN setting, even if the model remains certainly too unstable for correct distribution generation, adding additive skip-connections seems to stabilize the training a little since the colored bands are narrower than for the two other models. This suggests that some generator architectures may be better than others at learning mappings with large Lipschitz constants while staying stable.

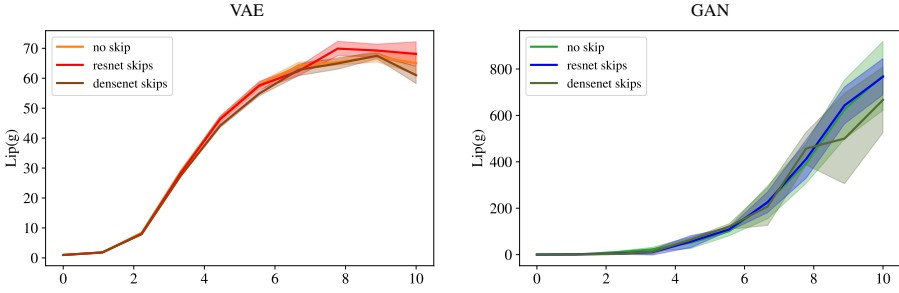

Figure 6: Evolution of the Lipschitz constant of the VAE decoder (left) and the GAN generator (right) trained on 50000 samples of $(1/2)[\mathrm{N}(-m, 1) + \mathrm{N}(m, 1)]$ for 3 different architectures of the generative network: simple feed-forward backbone, backbone with skip-connections of type "resnet", and backbone with skip-connections of type "densenet". Experiments are averaged over 5 runs and the colored bands correspond to +/- the standard deviation.

## 4.2 Experiments on MNIST

We train a VAE, a GAN and a SGM on two datasets derived from MNIST (LeCun et al., 1998): first, two images of two different digits (3 and 7) are chosen and 10000 noisy versions of theses images are drawn with a noise amount of $\sigma = 0.15$, forming a dataset of $n = 20002$ independent samples drawn from a balanced mixture of two Gaussian distributions in dimension $784 = 28 \times 28$. Second, we train the models on the subset of all 3 and 7 of MNIST. We emphasize that our goal is not reach state-of-the-art performance on this problem but rather to illustrate our theoretical results in a moderate dimensional setting.

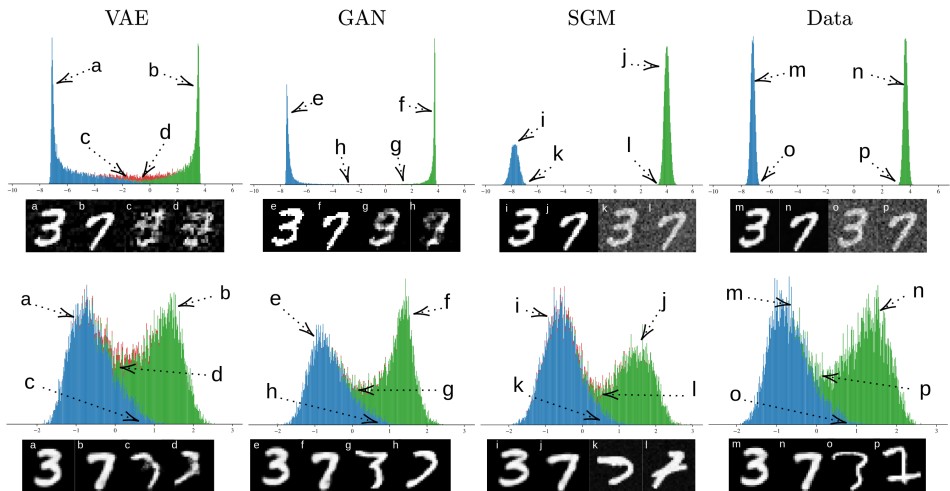

Figure 7: mixture of Gaussians (top): histograms of projections on the line passing through the mean of each Gaussian. Subset of MNIST (bottom): histograms of projections on the line passing through the barycenters of all the 3 and 7 in the deep Wasserstein embedding space. Bins of data are colored in blue if they are classified as 3, in green if classified as 7, and in red if classified as another digit.

**Mixture of Gaussians.** For this experiment, we set the dimension of the latent space in the GAN and the VAE to $784 = 28 \times 28$ since it is the intrinsic dimension of the support of the data distribution. In order to visualize the interpolation between modes, we project the data on the line passing through the mean of each Gaussian, i.e. the two original clean images, and we plot histograms of the one-dimensional projections. In order to understand which bins of data in the histograms correspond to which digit, we train a classifier and we assign a color in function of which digit the data have been classified as. Results can be found in Figure 7 top. Moreover, GAN and VAE both fail to generate noisy versions of the images. As in the univariate case, the SGM is able to not interpolate between

modes and seem to retrieve the Gaussian structure of the modes. This suggests that while direct push-forward models fail at representing multimodal distributions, considering stacked models with noise input at each step (as in SGM) might help to close the gap between the generated and the data distributions. However SGM does not manage to retrieve the right modes proportions. This is a well-known shortcoming of score-based models which has been studied in (Wenliang and Kanagawa, 2020).

**Subset of MNIST.** Finally we train the three different models on the subset of MNIST composed of all 3 and 7 (no Gaussian noise was added). We choose a latent dimension of 20 for the VAE and the GAN. Since the Euclidean distance is not a meaningful metric to compare the different digits of MNIST, we use the deep Wasserstein embedding proposed by Courty et al. (2018): an autoencoder is learned in a supervised fashion such that the Euclidean distance in the latent space approximates the Wasserstein distance between pairs of images of MNIST. In the learned Wasserstein space, we project data on the line passing through the Euclidean barycenters of all 3 and 7 and plot histograms of projections, using the same classifier as before. Results can be found in Figure 7 (bottom). Note that the distribution does not exhibit strong multimodality features contrary to the mixture of Gaussians settings, see Figure 7. As before, the VAE interpolate between modes, the GAN manages to not interpolate but generate a narrower histogram, and the score-based model does not interpolate and seems to recover the structure of the distribution, but doesn't retrieve the right modes proportions. However, we emphasize that all these models seem to perform better than on the previous dataset. A possible explanation of this is that the modes are less separated than in the Gaussian mixtures and therefore the model is easier to train.

## 5  Discussion

In this work, given a Lipschitz mapping $g$ and a measure $\nu$, we derived lower bounds on the total variation distance and the Kullback-Leibler divergence between the push-forward measure $g_{\#}\mu_p$ and $\nu$ depending on the Lipschitz constant of the mapping $g$. These bounds indicate how the mass between the modes of the push-forward measure depends on the Lipschitz constant of the push-forward mapping. They highlight the trade-off between the ability of VAEs and GANs to fit multimodal distributions and the stability of their training.

A common assumption in the imaging literature, validated empirically by Pope et al. (2020), is that distributions of natural images live on low dimensional manifolds. Understanding whether these distributions are composed of separated modes or not remains, to the best of our knowledge, an open problem. To that extent, the fact that unsupervised push-forward generative models perform well on datasets such as CelebA (Liu et al., 2015) could possibly be, in regard of our work, an indicator that the data distributions of those datasets are unimodal, or at least not composed of well separated modes.

Several techniques have been proposed in the literature to fit data distributions on disconnected manifolds. Most of them consist in overparametrizing the model, either by using stacked generative networks (Khayatkhoei et al., 2018; Mehr et al., 2019) or by learning a more complex latent distribution than the standard Gaussian (Gurumurthy et al., 2017; Rezende and Mohamed, 2015; Kingma et al., 2016; Luise et al., 2020). Other methods consist in rejecting a posteriori samples associated to large values of the Jacobian generator (Tanielian et al., 2020; Issenhuth et al., 2020). In this work, we empirically showed that score-based models seemed to be able to fit separated manifolds without model overparametrization or additional posterior sample rejection scheme. This suggests that the structure of the generation dynamic in these models is particularly adapted to (indirectly) learn mappings with large Lipschitz constants. Their good performance on multimodal distributions might follow from the fact that these models do not optimize directly the push-forward mapping itself and/or that noise is injected at each step during the generation process. Hence, a future perspective of work would be to study what are the structural aspects of diffusion models that play a significant role in their expressivity.

A possible limitation of this work is that the bounds derived on the Kullback-Leibler divergence and total variation distance are not tight (see Appendix S6), mainly because they take no account of the fact that when interpolating, $g_{\#}\mu_p$ has automatically less mass than $\nu$ on the modes since a significant amount of its total mass is between them. In future work, we plan to tighten the gap between our bounds and the true distance.

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
