# S1 Organization of the supplementary

The supplementary is organized as follows. First, in Appendix S2, we give details on why the generated distribution in each aforementioned generative model is a push-forward of a standard Gaussian distribution. In Appendix S3, we give the full proofs of all the theoretical results of the paper. In Appendix S4, we show a generalization of Corollary 5 in the case there are more than two disconnected manifolds. In Appendix S5, we give details on the experiments. Finally, in Appendix S6, we provide additional experimental results and additional visualizations of histograms of generated distributions for the univariate case, and generated data for the experiments on MNIST.

# S2 More details on push-forward generative models

In this section, we specify why in each aforementioned model the generated distribution is of the form $g_{\theta\#}\mu_p$ with $\mu_p = \mathrm{N}(0, \mathrm{Id}_p)$ being the standard Gaussian distribution in dimension $p$ and $g_\theta : \mathbb{R}^p \to \mathbb{R}^d$ being a deterministic mapping of parameter $\theta$.

## S2.1 Direct push-forward models

In GANs, the generated distribution is trivially of the form $g_{\theta\#}\mu_p$, where $g_\theta$ is the generator and $p$ is the dimension of the latent space. This is also the case in most of normalizing flow models, where $g_\theta$ is a composition of neural flows and $p$ is automatically equal to $d$ since the networks must be invertible. In the Gaussian-VAE model described in Kingma and Welling (2014), the conditional probability $p_\theta(x|z)$, with $z \in \mathbb{R}^p$ and $x \in \mathbb{R}^d$ being respectively the latent and observable variables, is of the form $p_\theta(x|z) = \mathrm{N}(f_\theta(z), h_\theta^2(z) \mathrm{Id}_d)$, where $f_\theta(z)$ and $g_\theta)$ are the outputs of the decoder. This is often simplified in practice to $p_\theta(x|z) = \mathrm{N}(f_\theta(z), c^2 \mathrm{Id}_d)$ with $c > 0$ being an hyperparameter of the model. Thus the generated distribution in Gaussian-VAE is of the form $g_{\#}\mu_p$, where $g_\theta : \mathbb{R}^{p+d} \to \mathbb{R}^d$ is the neural network defined as

$$g_\theta(z, z') = \begin{cases} f_\theta(z) + h_\theta^2(z)z' & \text{if } p_\theta(x|z) = \mathrm{N}(f_\theta(z), h_\theta^2(z) \mathrm{Id}_d) \\ f_\theta(z) + c^2 z' & \text{if } p_\theta(x|z) = \mathrm{N}(f_\theta(z), c^2 \mathrm{Id}_d) . \end{cases}$$

Moreover, if relaxing the conditional probability to $p_\theta(x|z) = \mathrm{N}(f_\theta(z), h_\theta^2(z) \mathrm{Id}_d)$ or $p_\theta(x|z) = \mathrm{N}(f_\theta(z), c^2 \mathrm{Id}_d)$ instead of $\delta_{f_\theta(z)}$ is crucial for the training of VAEs, this is not particularly relevant during inference since the relaxed conditional distribution is simply a noisy version of the push-forward measure $f_{\theta\#}\mu_p$. For this reason, the inference in VAEs is often done by simply sampling from the single output $f_\theta(z)$ of the decoder, with $z \sim \mu_p$, and so the generated distribution is trivially of the form $f_{\theta\#}\mu_p$ in that case.

## S2.2 Score-based generative models

In diffusion models, the generation process is an Euler-Maruyama discretization of the reverse-time denoising diffusion (Song et al., 2020)

$$\mathrm{d}X_t = \left( f(X_t, t) - g^2(t)\nabla \log p_t(X_t) \right) \mathrm{d}t + g(t)\mathrm{d}B_t ,$$

where $(X_t)_{t \in [0,T]}$ is a random process on $\mathbb{R}^d$, $(B_t)_{t \in [0,T]}$ is a Brownian motion, and where $f(.,t) : \mathbb{R}^d \to \mathbb{R}^d$ is a vector-valued function called the drift operator, $g : \mathbb{R} \to \mathbb{R}$ is a real-valued function called the diffusion coefficient and $\nabla \log p_t(X_t)$ is the score of the marginal law of $X_t$, which is approximated by a neural network $s_\theta(X_t, t)$. This discretization yields for instance, for an appropriate choice of $f$ and $g$, to the annealed Langevin dynamic (Song and Ermon, 2019):

$$\begin{cases} x_0 = z_0 & (z_0 \sim \mathrm{N}(0, \mathrm{Id}_d)) \\ x_{k+1} = x_k + (\alpha_k/2)s_\theta(x_k, \sigma_k) + \sqrt{\alpha_k}z_{k+1} & (z_{k+1} \sim \mathrm{N}(0, \mathrm{Id}_d)) , \end{cases}$$

where $\alpha_k = \varepsilon\sigma_k^2/\sigma_0^2$ with $\varepsilon > 0$ being an hyperparameter of the model and $(\sigma_k)_{k\geq 0}$ is such that it exists $K$ (another hyperparameter of the model) such that $(\sigma_{Ki})_{i\geq 0}$ is a geometric progression, and for all $k \geq 0$, $\sigma_k = \sigma_{K\lfloor k/K \rfloor}$. Denoting for $k \geq 1$, $h_\theta^k : \mathbb{R}^{2d} \to \mathbb{R}^d$ the function defined as

$$h_\theta^k(x, y) = x + (\alpha_k/2)s_\theta(x, \sigma_k) + \sqrt{\alpha_k}y ,$$

it follows that when the data are generated with a Langevin dynamic of $N$ iterations, the generated distribution is of the form $g_{\theta\#}\mu_p$ with $p = d(N + 1)$ and where $g_\theta : \mathbb{R}^{d(N+1)} \to \mathbb{R}^d$ is the function defined as

$$g_\theta(z_0, z_1, z_2, \ldots, z_{N-1}, z_N) = h_\theta^N \left( h_\theta^{N-1} \left( \ldots \left( h_\theta^2 \left( h_\theta^1(z_0, z_1), z_2 \right), \ldots \right), z_{N-1} \right), z_N \right) .$$

## S3 Proofs of the theoretical results

### S3.1 Proof of Theorem 1

We start by recalling the Gaussian isoperimetric inequality Sudakov and Tsirelson (1978).

**Lemma S9.** *Let* $\mathsf{A} \in \mathcal{B}(\mathbb{R}^p)$ *and* $\mu_p = \mathrm{N}(0, \mathrm{Id}_p)$*. Then we have*

$$\mu_p^+(\partial \mathsf{A}) \geq \varphi(\Phi^{-1}(\mu_p(\mathsf{A}))) ,$$

*where* $\varphi(x) = (2\pi)^{-1/2} \exp[-x^2/2]$ *and* $\Phi(x) = \int_{-\infty}^x \varphi(t)\mathrm{d}t$*. Equivalently, for all* $r \geq 0$

$$\mu_p(\mathsf{A}_r) \geq \Phi(r + \Phi^{-1}(\mu_p(\mathsf{A}))) .$$

In particular, using Lemma S9, one can show that among all sets of given Gaussian measure $\mu_p$, half-spaces have the minimal $\mu_p$-surface area. We are now ready to turn to the proof of Theorem 1.

*Proof of Theorem 1.* Let $\mathsf{A} \in \mathcal{B}(\mathbb{R}^d)$ such that $g_{\#}\mu_p(\mathsf{A}) > 0$ (note that if $g_{\#}\mu_p(\mathsf{A}) = 0$ then the result is trivial). First, we show that for any $\varepsilon > 0$, $g((g^{-1}(\mathsf{A}))_{\varepsilon/\mathrm{Lip}(g)}) \subset \mathsf{A}_\varepsilon$. Let $x$ be in $g((g^{-1}(\mathsf{A}))_{\varepsilon/\mathrm{Lip}(g)})$. There exists $z_1 \in (g^{-1}(\mathsf{A}))_{\varepsilon/\mathrm{Lip}(g)}$ such that $g(z_1) = x$. There also exists $z_2 \in g^{-1}(\mathsf{A})$ such that

$$\|z_1 - z_2\| \leq \varepsilon/\mathrm{Lip}(g) .$$

Hence, we have that

$$\|x - a\| \leq \mathrm{Lip}(g)\|z_1 - z_2\| \leq \varepsilon ,$$

where $a = g(z_2)$. Since $z_2 \in g^{-1}(\mathsf{A})$, $a \in \mathsf{A}$, and therefore $x \in \mathsf{A}_\varepsilon$. Using this result, the fact that $g_{\#}\mu_p(\mathsf{B}) = \mu_p(g^{-1}(\mathsf{B}))$ and $\mathsf{B} \subset g^{-1}(g(\mathsf{B}))$ for any $\mathsf{B} \in \mathcal{B}(\mathbb{R}^d)$, we have

$$\liminf_{\varepsilon \to 0^+}\{g_{\#}\mu_p(\mathsf{A}_\varepsilon) - g_{\#}\mu_p(\mathsf{A})\}/\varepsilon \geq \liminf_{\varepsilon \to 0^+}\{g_{\#}\mu_p(g((g^{-1}(\mathsf{A}))_{\varepsilon/\mathrm{Lip}(g)})) - g_{\#}\mu_p(\mathsf{A})\}/\varepsilon$$
$$\geq \liminf_{\varepsilon \to 0^+}\{\mu_p((g^{-1}(\mathsf{A}))_{\varepsilon/\mathrm{Lip}(g)}) - \mu_p(g^{-1}(\mathsf{A}))\}/\varepsilon . \quad \text{(S1)}$$

Using Lemma S9, we have

$$\mathrm{Lip}(g)\liminf_{\varepsilon \to 0^+}\{(\mu_p((g^{-1}(\mathsf{A}))_{\varepsilon/\mathrm{Lip}(g)}) - \mu_p(g^{-1}(\mathsf{A})))\}/\varepsilon \geq \varphi(\Phi^{-1}(\mu_p(g^{-1}(\mathsf{A})))) ,$$

Combining this result and (S1), we get that

$$\mathrm{Lip}(g)(g_{\#}\mu_p)^+(\partial \mathsf{A}) \geq \varphi(\Phi^{-1}(g_{\#}\mu_p(\mathsf{A}))) .$$

In addition, using Lemma S9, we have for all $r \geq 0$

$$\mu_p((g^{-1}(\mathsf{A}))_{r/\mathrm{Lip}(g)}) \geq \Phi(r/\mathrm{Lip}(g) + \Phi^{-1}(\mu_p(g^{-1}(\mathsf{A})))) .$$

Using this result and that $g((g^{-1}(\mathsf{A}))_{r/\mathrm{Lip}(g)}) \subset \mathsf{A}_r$, we have for any $r \geq 0$

$$g_{\#}\mu_p(\mathsf{A}_r) = \mu_p(g^{-1}(\mathsf{A}_r)) \geq \mu_p((g^{-1}(\mathsf{A}))_{r/\mathrm{Lip}(g)}) \geq \Phi(r/\mathrm{Lip}(g) + \Phi^{-1}(g_{\#}\mu_p(\mathsf{A}))) .$$

$\square$

## S3.2 Proof of Corollary 2

We prove the corollary when $\nu = \lambda \mathrm{N}(-m, \sigma^2 \,\mathrm{Id}_d) + (1-\lambda)\mathrm{N}(m, \sigma^2 \,\mathrm{Id}_d)$ since the problem can always be reduced to that case by translation and setting $m = (m_2 - m_1)/2$. Let $\mathsf{H}$ be defined by $\mathsf{H} = \{x \in \mathbb{R}^d | m^T x \geq 0\}$. Note that for any $x \in \partial \mathsf{H}$, $\|x - m\| = \|x + m\|$. Since the problem is invariant by rotation, we can consider without any loss of generality that $m = (\|m\|, 0, \ldots, 0)$. In that case, we have $\nu = \nu_1 \otimes \mathrm{N}(0, \sigma^2 \,\mathrm{Id}_{d-1})$, where $\nu_1 = \lambda \mathrm{N}(-\|m\|, \sigma^2) + (1-\lambda)\mathrm{N}(\|m\|, \sigma^2)$, and $\otimes$ is the tensor product between measures. In this case, we have that $\mathsf{H} = \{x_1 \geq 0\} \times \mathbb{R}^{d-1}$. Therefore, we have

$$\nu^+(\partial \mathsf{H}) = \liminf_{\varepsilon \to 0^+}\{(\textstyle\int_{\mathsf{H}_\varepsilon} p_\nu(x)\mathrm{d}x - \int_{\mathsf{H}} p_\nu(x)\mathrm{d}x)\}/\varepsilon\ ,$$
$$= \liminf_{\varepsilon \to 0^+}\{(\textstyle\int_{-\varepsilon}^{+\infty} \int_{\mathbb{R}^{d-1}} p_{\nu_1}(x_1)h(y)\mathrm{d}x_1\mathrm{d}y - \int_0^{+\infty} \int_{\mathbb{R}^{d-1}} p_{\nu_1}(x_1)h(y)\mathrm{d}x_1\mathrm{d}y)\}/\varepsilon\ ,$$

where $p_\nu$ and $p_{\nu_1}$ are the respective densities of $\nu$ and $\nu_1$, and $h$ is the density of $\mathrm{N}(0, \sigma^2 I_{d-1})$. It follows that

$$\nu^+(\partial \mathsf{H}) = \liminf_{\varepsilon \to 0^+}(1/\varepsilon) \textstyle\int_{-\varepsilon}^0 p_{\nu_1}(x_1)(\int_{\mathbb{R}^{d-1}} h(y)\mathrm{d}y)\mathrm{d}x_1$$
$$= \liminf_{\varepsilon \to 0^+}(1/\varepsilon) \textstyle\int_{-\varepsilon}^0 p_{\nu_1}(x_1)\mathrm{d}x_1 = p_{\nu_1}(0) = (2\pi\sigma^2)^{-1/2} \exp[-\|m\|^2/(2\sigma^2)]\ .$$

Applying Theorem 1, we get that

$$\mathrm{Lip}(g) \geq \varphi(\Phi^{-1}(\nu(\mathsf{H})))/\nu^+(\partial \mathsf{H})\ .$$

Furthermore, one can derive that

$$\nu(\mathsf{H}) = \lambda(1 - \Phi(m/\sigma)) + \Phi(m/\sigma)(1 - \lambda)$$
$$= \lambda(1 - 2\Phi(m/\sigma)) + \Phi(m/\sigma)\ .$$

Observing that $\lambda - \nu(\mathsf{H})$ is an increasing function of $\lambda$ and $\lambda - \nu(\mathsf{H}) = 0$ if $\lambda = 1/2$, we get that $\lambda \leq \nu(\mathsf{H})$ if $\lambda \leq 1/2$ and $\lambda \geq \nu(\mathsf{H})$ if $\lambda \geq 1/2$. Since $\varphi \circ \Phi^{-1}$ reaches its maximum in $1/2$, it follows that for any $\lambda \in (0, 1)$ we have

$$\varphi(\Phi^{-1}(\nu(\mathsf{H}))) \geq \varphi(\Phi^{-1}(\lambda))\ ,$$

and thus

$$\mathrm{Lip}(g) \geq (2\pi)^{1/2}\sigma\varphi(\Phi^{-1}(\lambda)) \exp[\|m\|^2/(2\sigma^2)]$$
$$\geq \sigma \exp[\|m\|^2/(2\sigma^2) - (\Phi^{-1}(\lambda))^2/2]\ ,$$

which concludes the proof.

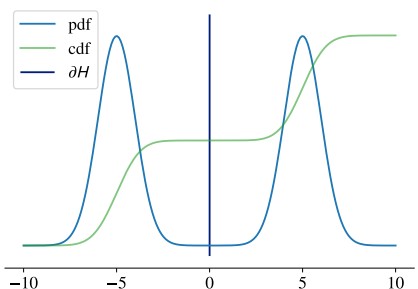

Figure S1: The hypersurface $\partial \mathsf{H}$ in the univariate case.

## S3.3 Proof of Corollary 3

Since $\nu$ admits a density $p_\nu$ with respect to the Lesbegue measure, it follows that $\Phi_\nu$ is absolutely continuous and therefore differentiable almost everywhere w.r.t. the Lebesgue measure using the Lebesgue differentiation theorem. Moreover, since $\mathrm{supp}(\nu) = \mathbb{R}$, it follows that $\Phi_\nu : \mathbb{R} \to (0, 1)$ is

increasing and therefore is bijective, and so $T_{\mathrm{OT}} = \Phi_\nu^{-1} \circ \Phi$ is also differentiable almost everywhere w.r.t. the Lebesgue measure and bijective, with inverse $T_{\mathrm{OT}}^{-1} = \Phi^{-1} \circ \Phi_\nu$, using (Peyré and Cuturi, 2019, Remark 2.29). Therefore, for any $x \in \mathbb{R}$ we have

$$T_{\mathrm{OT}}'(x) = \varphi(x)/p_\nu(T_{\mathrm{OT}}(x))$$
$$= \varphi(\Phi^{-1}(\Phi_\nu(T_{\mathrm{OT}}(x))))/p_\nu(T_{\mathrm{OT}}(x)) .$$

Let $y \in \mathbb{R}$. Using Theorem 1 with $\mathsf{A} = (-\infty, y]$ we get that for any $g : \mathbb{R}^p \to \mathbb{R}$, Lipschitz such that $g_\# \mu_p = \nu$,

$$\mathrm{Lip}(g) \geq \sup_{y \in \mathbb{R}} \varphi(\Phi^{-1}(\Phi_\nu(y)))/p_\nu(y) ,$$

and so, since $T_{\mathrm{OT}}$ is bijective

$$\mathrm{Lip}(g) \geq \sup_{x \in \mathbb{R}} |T_{\mathrm{OT}}'(x)| ,$$

which concludes the proof.

### S3.4   Proof of Theorem 4

Let $\mathsf{A} \in \mathcal{B}(\mathbb{R}^d)$ and let $r > 0$. We have on one hand

$$|g_\# \mu_p(\mathsf{A}_r \setminus \mathsf{A})| \leq |g_\# \mu_p(\mathsf{A}_r \setminus \mathsf{A}) - \nu(\mathsf{A}_r \setminus \mathsf{A})| + |\nu(\mathsf{A}_r \setminus \mathsf{A})|$$
$$\leq d_{\mathrm{TV}}(g_\# \mu_p, \nu) + \nu(\mathsf{A}_r \setminus \mathsf{A}) .$$

Using Theorem 1, we get

$$|g_\# \mu_p(\mathsf{A}_r \setminus \mathsf{A})| = g_\# \mu_p(\mathsf{A}_r) - g_\# \mu_p(\mathsf{A}) \geq \Phi\left(r/\mathrm{Lip}(g) + \Phi^{-1}(g_\# \mu_p(\mathsf{A}))\right) - g_\# \mu_p(\mathsf{A}) ,$$

and so

$$d_{\mathrm{TV}}(g_\# \mu_p, \nu) \geq \alpha_g(\mathsf{A}, r) - g_\# \mu_p(\mathsf{A}) - \nu(\mathsf{A}_r \setminus \mathsf{A}) ,$$

where $\alpha_g(\mathsf{A}, r) = \Phi\left(r/\mathrm{Lip}(g) + \Phi^{-1}(g_\# \mu_p(\mathsf{A}))\right)$. On the other hand, we have

$$|g_\# \mu_p(\mathsf{A}_r)| \leq |g_\# \mu_p(\mathsf{A}_r) - \nu(\mathsf{A}_r)| + |\nu(\mathsf{A}_r)|$$
$$\leq d_{\mathrm{TV}}(g_\# \mu_p, \nu) + \nu(\mathsf{A}_r \setminus \mathsf{A}) + \nu(\mathsf{A}) .$$

Using Theorem 1, we get

$$|g_\# \mu_p(\mathsf{A}_r)| \geq \Phi\left(r/\mathrm{Lip}(g) + \Phi^{-1}(g_\# \mu_p(\mathsf{A}))\right) ,$$

and so

$$d_{\mathrm{TV}}(g_\# \mu_p, \nu) \geq \alpha_g(\mathsf{A}, r) - \nu(\mathsf{A}) - \nu(\mathsf{A}_r \setminus \mathsf{A}) ,$$

which concludes the proof.

### S3.5   Proof of Corollary 5

To prove Corollary 5, we will need the following lemma:

**Lemma S10.** *Let $\mathsf{A} \in \mathcal{B}(\mathbb{R}^d)$ and $r > 0$. We denote $\mathsf{B} = (\mathsf{A}_r)^c$. Then*

$$\mathsf{B}_r \subset \bar{\mathsf{A}^c} ,$$

*where $\bar{\mathsf{A}^c}$ denotes the closure of the complementary of $\mathsf{A}$.*

*Proof.* Let $x \in \mathsf{B}_r$. There exists $b \in \mathsf{B}$ such that $\|x - b\| \leq r$. Moreover, since $\mathsf{B} = (\mathsf{A}_r)^c$, it follows that for all $a \in \mathsf{A}$,

$$\|b - a\| > r .$$

Then

$$r < \|b - x\| + \|x - a\| ,$$

and so, it follows that for all $a \in \mathsf{A}$,

$$\|x - a\| > 0 .$$

Thus $x \in \bar{\mathsf{A}^c}$. $\qquad\square$

Now we are ready to turn to the proof of Corollary 5.

*Proof of Corollary 5.* We set $r = d(\mathsf{M}_1, \mathsf{M}_2)/2$ and $\mathsf{A} = (\mathsf{M}_1)_r$. Using Theorem 4, we have

$$d_{\mathrm{TV}}(g_\#\mu_p, \nu) \geq \alpha_g(\mathsf{A}, r) - \min\{g_\#\mu_p(\mathsf{A}), \nu(\mathsf{A})\} - \nu(\mathsf{A}_r \setminus \mathsf{A}) \;.$$

First we suppose that $g_\#\mu_p(\mathsf{A}) \geq \nu(\mathsf{A})$: since $\Phi$ is a non-decreasing function, it follows that

$$\alpha_g(\mathsf{A}, r) = \Phi\left(r/\mathrm{Lip}(g) + \Phi^{-1}(g_\#\mu_p(\mathsf{A}))\right) \geq \Phi\left(r/\mathrm{Lip}(g) + \Phi^{-1}(\nu(\mathsf{A}))\right) \;.$$

Moreover $\min\{g_\#\mu_p(\mathsf{A}), \nu(\mathsf{A})\} = \nu(\mathsf{A}) = \lambda = \Phi(\Phi^{-1}(\lambda))$ and so it follows

$$d_{\mathrm{TV}}(g_\#\mu_p, \nu) \geq \Phi\left(d(\mathsf{M}_1, \mathsf{M}_2)/(2\mathrm{Lip}(g)) + \Phi^{-1}(\lambda)\right) - \Phi(\Phi^{-1}(\lambda)) \geq \int_{\Phi^{-1}(\lambda)}^{r/\mathrm{Lip}(g)+\Phi^{-1}(\lambda)} \varphi(t)dt \;,$$

since $\nu$ has no mass on $\mathsf{A}_r \setminus \mathsf{A}$. Now we suppose that $g_\#\mu_p(\mathsf{A}) \leq \nu(\mathsf{A})$: we then set $\mathsf{B} = \mathsf{A}^c$. Since $g_\#\mu_p(\mathsf{A}) \leq \nu(\mathsf{A})$, we have $g_\#\mu_p(\mathsf{B}) \geq \nu(\mathsf{B})$. Applying Theorem 4, and the same reasoning as before we get

$$d_{\mathrm{TV}}(g_\#\mu_p, \nu) \geq \alpha_g(\mathsf{B}, r) - \min\{g_\#\mu_p(\mathsf{B}), \nu(\mathsf{B})\} - \nu(\mathsf{B}_r \setminus \mathsf{B})$$
$$\geq \Phi\left(d(\mathsf{M}_1, \mathsf{M}_2)/(2\mathrm{Lip}(g)) + \Phi^{-1}(1-\lambda)\right) - \Phi(\Phi^{-1}((1-\lambda)) - \nu(\mathsf{B}_r \setminus \mathsf{B}) \;.$$

Using Lemma S10, we get that $\nu(\mathsf{B}_r \setminus \mathsf{B}) \leq \nu(\bar{\mathsf{A}}^c \setminus (\mathsf{A}_r)^c)$ but $\nu(\bar{\mathsf{A}}^c \setminus (\mathsf{A}_r)^c) = 0$ since $\nu$ has no mass on $\bar{\mathsf{A}}^c \setminus (\mathsf{A}_r)^c$ except on its boundary and so its follows that

$$d_{\mathrm{TV}}(g_\#\mu_p, \nu) \geq \Phi\left(d(\mathsf{M}_1, \mathsf{M}_2)/(2\mathrm{Lip}(g)) + \Phi^{-1}(1-\lambda)\right) - \Phi(\Phi^{-1}((1-\lambda))$$
$$\geq \Phi\left(d(\mathsf{M}_1, \mathsf{M}_2)/(2\mathrm{Lip}(g)) - \Phi^{-1}(\lambda)\right) - \Phi(-\Phi^{-1}(\lambda))$$
$$\geq \int_{-\Phi^{-1}(\lambda)}^{r/\mathrm{Lip}(g)-\Phi^{-1}(\lambda)} \varphi(t)dt \;,$$

since $\Phi^{-1}(1-\lambda) = -\Phi^{-1}(\lambda)$. Since $\lambda \geq 1/2$, it follows that $\Phi^{-1}(\lambda) \geq 0$ and so

$$\int_{-\Phi^{-1}(\lambda)}^{r/\mathrm{Lip}(g)-\Phi^{-1}(\lambda)} \varphi(t)dt \geq \int_{\Phi^{-1}(\lambda)}^{r/\mathrm{Lip}(g)+\Phi^{-1}(\lambda)} \varphi(t)dt \;,$$

which concludes the proof. $\qquad \square$

## S3.6 Proof of Corollary 6

As previously, we prove the corollary when $\nu = (1/2)[\mathrm{N}(-m, \sigma^2 \mathrm{Id}_d) + \mathrm{N}(m, \sigma^2 \mathrm{Id}_d)]$ since the problem can always be reduced to that case by translation and setting $m = (m_2 - m_1)/2$. Since the problem is invariant by rotation, we can assume without any loss of generality that $m = (\|m\|, 0, \ldots, 0)$. Let $\mathsf{H}$ be the half-space of $\mathbb{R}^d$ defined by $\mathsf{H} = (-\infty, 0] \times \mathbb{R}^{d-1}$ and we set $r = \|m\|/2\sigma$. First we suppose that $g_\#\mu_p(\mathsf{H}) \geq \nu(\mathsf{H})$: using Theorem 4, we get that

$$d_{\mathrm{TV}}(g_\#\mu_p, \nu) \geq \alpha_g(\mathsf{H}, r) - \min\{g_\#\mu_p(\mathsf{H}), \nu(\mathsf{H})\} - \nu(\mathsf{H}_r \setminus \mathsf{H}) \;,$$

with $\mathsf{H}_r = (-\infty, \|m\|/2\sigma] \times \mathbb{R}^{d-1}$. On one hand we have that $\nu = \nu_1 \otimes \mathrm{N}(0, \sigma^2 \mathrm{Id}_{d-1})$, where $\nu_1 = (1/2)[\mathrm{N}(-\|m\|, \sigma^2) + \mathrm{N}(\|m\|, \sigma^2)]$ and so $\nu(\mathsf{H}_r \setminus \mathsf{H}) = \nu_1([0, \|m\|/2\sigma])$. On the other hand we have that $\min\{g_\#\mu_p(\mathsf{H}), \nu(\mathsf{H})\} = \nu(\mathsf{H})$ and $g_\#\mu_p(\mathsf{H}) \geq 1/2$ since $g_\#\mu_p(\mathsf{H}) \geq \nu(\mathsf{H})$. Hence it follows that

$$d_{\mathrm{TV}}(g_\#\mu_p, \nu) \geq \Phi(r/\mathrm{Lip}(g)) - 1/2 - \nu_1([0, \|m\|/2\sigma]) \;.$$

Now we suppose that $g_\#\mu_p(\mathsf{H}) \leq \nu(\mathsf{H})$: we then set $\mathsf{H}_2 = (0, +\infty] \times \mathbb{R}^{d-1}$. Since $g_\#\mu_p(\mathsf{H}) \leq 1/2$, we get that $g_\#\mu_p(\mathsf{H}_2) \geq 1/2$ and so $g_\#\mu_p(\mathsf{H}_2) \geq \nu(\mathsf{H}_2)$. Hence we retrieve the previous case and so it follows that

$$d_{\mathrm{TV}}(g_\#\mu_p, \nu) \geq \Phi(r/\mathrm{Lip}(g)) - 1/2 - \nu_1([-\|m\|/2\sigma, 0]) \;.$$

Since $\nu_1([-\|m\|/2\sigma, 0]) = \nu_1([0, \|m\|/2\sigma])$, we get in both cases

$$d_{\mathrm{TV}}(g_\#\mu_p, \nu) \geq \Phi(r/\mathrm{Lip}(g)) - 1/2 - \nu_1([0, \|m\|/2\sigma]) \;.$$

Now we derive the value of $\nu_1([0, \|m\|/2\sigma])$:

$$\nu_1([0, \|m\|/2\sigma]) = (1/2) \int_0^{m/2\sigma} (2\pi\sigma^2)^{-1/2} \exp[-(x+m)^2/2\sigma^2]dx$$
$$+ (1/2) \int_0^{m/2\sigma} (2\pi\sigma^2)^{-1/2} \exp[-(x-m)^2/2\sigma^2]dx$$
$$= (1/2) \int_{-m/2\sigma}^{m/2\sigma} (2\pi\sigma^2)^{-1/2} \exp[-(x+m)^2/2\sigma^2]dx$$
$$= (1/2) \int_{\|m\|(2\sigma-1)/2\sigma^2}^{\|m\|(2\sigma+1)/2\sigma^2} \varphi(x)dx \;,$$

which concludes the proof.

## S3.7 Proof of Theorem 7

Let $\mathsf{A} \in \mathcal{B}(\mathbb{R}^d)$, $r > 0$ and $\zeta > 0$. We set for any $x \in \mathbb{R}^d$ $f(x) = \zeta \chi_{\mathsf{A}_r \setminus \mathsf{A}}(x)$, where $\chi_{\mathsf{A}}$ denotes the characteristic function of the set $\mathsf{A}$. Since $f$ is bounded, it follows that

$$d_{KL}(g_\#\mu_p || \nu) \geq \int_{\mathbb{R}^d} f(x) dg_\#\mu_p(x) - \log\left(\int_{\mathbb{R}^d} e^{f(x)} d\nu(x)\right)$$
$$\geq \zeta g_\#\mu_p(\mathsf{A}_r \setminus \mathsf{A}) - \log\left(1 + (e^\zeta - 1)\nu(\mathsf{A}_r \setminus \mathsf{A})\right).$$

Using Theorem 1, we get

$$g_\#\mu_p(\mathsf{A}_r \setminus \mathsf{A}) = g_\#\mu_p(\mathsf{A}_r) - g_\#\mu_p(\mathsf{A}) \geq \Phi\left(r/\mathrm{Lip}(g) + \Phi^{-1}(g_\#\mu_p(\mathsf{A}))\right) - g_\#\mu_p(\mathsf{A}).$$

Thus we get

$$d_{\mathrm{KL}}(g_\#\mu_p || \nu) \geq \sup\{J(\zeta, \mathsf{A}, r) : \zeta \in \mathbb{R}, \mathsf{A} \in \mathcal{B}(\mathbb{R}^d), r > 0\},$$

where the functional $J$ is defined by

$$J(\zeta, \mathsf{A}, r) = \zeta\left(\Phi\left(r/\mathrm{Lip}(g) + \Phi^{-1}(g_\#\mu_p(\mathsf{A}))\right) - g_\#\mu_p(\mathsf{A})\right)$$
$$-\log\left(1 + (e^\zeta - 1)\nu(\mathsf{A}_r \setminus \mathsf{A})\right).$$

Differentiating $J$ with respect to $\zeta$, we get that

$$\nabla_\zeta J(\zeta, \mathsf{A}, r) = \beta_g(\mathsf{A}, r) - (e^\zeta \nu(\mathsf{A}_r \setminus \mathsf{A}))/(1 + (e^\zeta - 1)\nu(\mathsf{A}_r \setminus \mathsf{A})),$$

where $\beta_g(\mathsf{A}, r) = \Phi\left(r/\mathrm{Lip}(g) + \Phi^{-1}(g_\#\mu_p(\mathsf{A}))\right) - g_\#\mu_p(\mathsf{A})$. Applying the first order condition, we get that:

$$\zeta^* = \log[\beta_g(\mathsf{A}, r)(1 - \nu(\mathsf{A}_r \setminus \mathsf{A}))] - \log[\nu(\mathsf{A}_r \setminus \mathsf{A})(1 - \beta_g(\mathsf{A}, r))].$$

By re-injecting the value of $\zeta^*$, we get

$$\zeta^* \beta_g(\mathsf{A}, r) - \log\left(1 + (e^{\zeta^*} - 1)\nu(\mathsf{A}_r \setminus \mathsf{A})\right) = \beta_g(\mathsf{A}, r) \log\left(\frac{\beta_g(\mathsf{A}, r)(1 - \nu(\mathsf{A}_r \setminus \mathsf{A}))}{\nu(\mathsf{A}_r \setminus \mathsf{A})(1 - \beta_g(\mathsf{A}, r))}\right)$$
$$- \log\left(\frac{1 - \nu(\mathsf{A}_r \setminus \mathsf{A})}{1 - \beta_g(\mathsf{A}, r)}\right)$$
$$= \beta_g(\mathsf{A}, r) \log\left(\frac{\beta_g(\mathsf{A}, r)}{\nu(\mathsf{A}_r \setminus \mathsf{A})}\right)$$
$$+ (1 - \beta_g(\mathsf{A}, r)) \log\left(\frac{1 - \beta_g(\mathsf{A}, r)}{1 - \nu(\mathsf{A}_r \setminus \mathsf{A})}\right),$$

which concludes the proof.

## S3.8 Proof of Corollary 8

As previously, we prove the corollary when $\nu = (1/2)[\mathrm{N}(-m, \sigma^2 \mathrm{Id}_d) + \mathrm{N}(m, \sigma^2 \mathrm{Id}_d)]$ since the problem can always be reduced to that case by translation and setting $m = (m_2 - m_1)/2$. Since the problem is invariant by rotation, we can assume without any loss of generality that $m = (\|m\|, 0, \ldots, 0)$. Furthermore, observe that the half-space $\{(m_2 - m_1)^T (x - (m_1 + m_2)/2) \leq 0 : x \in \mathbb{R}^d\}$ becomes $(-\infty, 0] \times \mathbb{R}^{d-1}$ in that case, and that the condition $\lambda \in (0, 1/2]$ is indeed non-restrictive since the problem is invariant by rotation. We set as before $\mathsf{H} = (-\infty, 0] \times \mathbb{R}^{d-1}$ and $r = \|m\|/2\sigma$.

Applying Theorem 7, we get

$$d_{\mathrm{KL}}(g_\#\mu_p || \nu) \geq \beta_g(\mathsf{H}, r) \log\left(\frac{\beta_g(\mathsf{H}, r)}{\nu(\mathsf{H}_r \setminus \mathsf{H})}\right) + (1 - \beta_g(\mathsf{H}, r)) \log\left(\frac{1 - \beta_g(\mathsf{H}, r)}{1 - \nu(\mathsf{H}_r \setminus \mathsf{H})}\right).$$

On one hand, we get

$$\beta_g(\mathsf{H}, r) = \Phi\left(r/\mathrm{Lip}(g) + \Phi^{-1}(g_\#\mu_p(\mathsf{H}))\right) - g_\#\mu_p(\mathsf{H})$$
$$= \Phi\left(r/\mathrm{Lip}(g) + \Phi^{-1}(g_\#\mu_p(\mathsf{H}))\right) - \Phi\left(\Phi^{-1}(g_\#\mu_p(\mathsf{H}))\right)$$
$$= \int_{\Phi^{-1}(\lambda)}^{\|m\|/2\sigma\mathrm{Lip}(g) + \Phi^{-1}(\lambda)} \varphi(t) dt$$
$$= \int_{-\Phi^{-1}(1-\lambda)}^{\|m\|/2\sigma\mathrm{Lip}(g) - \Phi^{-1}(1-\lambda)} \varphi(t) dt,$$

noting $\lambda = g_\# \mu_p(\mathsf{H})$. We replaced $\Phi^{-1}(\lambda)$ by $-\Phi^{-1}(1-\lambda)$ in order to emphasize that $\Phi^{-1}(\lambda) \leq 0$ since $\lambda \leq 1/2$. Observe that if we supposed $\lambda \geq 1/2$, we would have $\beta_g(\mathsf{H}^c, r) \geq \beta_g(\mathsf{H}, r)$ and so the bound that we would have found by reasoning on $\mathsf{H}$ would have been sub-optimal. On the other hand, observing as before that $\nu = \nu_1 \otimes \mathrm{N}(0, \sigma^2 \, \mathrm{Id}_{d-1})$, where $\nu_1 = (1/2)[\mathrm{N}(-\|m\|, \sigma^2) + \mathrm{N}(\|m\|, \sigma^2)]$, we get that

$$
\begin{aligned}
\nu(\mathsf{H}_r \setminus \mathsf{H}) &= \nu_1([0, \|m\|/2\sigma]) \\
&= (1/2) \int_{\|m\|(2\sigma-1)/2\sigma^2}^{\|m\|(2\sigma+1)/2\sigma^2} \varphi(t)\mathrm{d}t \, ,
\end{aligned}
$$

which concludes the proof.

## S4  Additional theoretical result

In this section we derive a generalization of Corollary 5 when $\nu$ is a distribution whose support is composed of more than two disconnected manifolds.

**Corollary S11.** *Let $\nu$ be a measure on $\mathbb{R}^d$ on $N$ disconnected manifolds $(\mathsf{M}_1, \ldots, \mathsf{M}_N)$, and let $g : \mathbb{R}^p \to \mathbb{R}^d$ be a Lipschitz function. Then,*

$$
d_{\mathrm{TV}}(g_\# \mu_p, \nu) \geq \max_{I \subset \llbracket 1, N \rrbracket} \int_{\Phi^{-1}(\lambda)}^{d(\bigsqcup_{i \in I} \mathsf{M}_i, \bigsqcup_{j \in \llbracket 1, N \rrbracket \setminus I} \mathsf{M}_j)/2\mathrm{Lip}(g) + \Phi^{-1}(\lambda)} \varphi(t)dt \, ,
$$

*where for $\mathsf{A}, \mathsf{B} \in \mathcal{B}(\mathbb{R}^d)$, $d(\mathsf{A}, \mathsf{B}) = \inf\{\|a - b\| \; : \; a \in \mathsf{A}, b \in \mathsf{B}\}$, and $\lambda = \nu\left(\bigsqcup_{i \in I} \mathsf{M}_i\right)$ if $\nu\left(\bigsqcup_{i \in I} \mathsf{M}_i\right) \geq 1/2$ and $\lambda = 1 - \nu\left(\bigsqcup_{i \in I} \mathsf{M}_i\right)$ otherwise.*

*Proof.* Let $I \subset \llbracket 1, N \rrbracket$. First, we suppose that $\nu\left(\bigsqcup_{i \in I} \mathsf{M}_i\right) \geq 1/2$. Since $\nu$ can be seen as a bi-modal distribution on the two disconnected sets $\bigsqcup_{i \in I} \mathsf{M}_i$ and $\bigsqcup_{j \in \llbracket 1, N \rrbracket \setminus I} \mathsf{M}_j$, we can apply Corollary 5. Thus we get

$$
d_{\mathrm{TV}}(g_\# \mu_p, \nu) \geq \int_{\Phi^{-1}(\lambda)}^{d(\bigsqcup_{i \in I} \mathsf{M}_i, \bigsqcup_{j \in \llbracket 1, N \rrbracket \setminus I} \mathsf{M}_j)/2\mathrm{Lip}(g) + \Phi^{-1}(\lambda)} \varphi(t)dt \, .
$$

If $\nu\left(\bigsqcup_{i \in I} \mathsf{M}_i\right) \leq 1/2$, we can still apply Corollary 5 by interchanging the roles of $\bigsqcup_{i \in I} \mathsf{M}_i$ and $\bigsqcup_{j \in \llbracket 1, N \rrbracket \setminus I} \mathsf{M}_j$, thus we get also Inequality (S4) in that case, which concludes the proof. □

## S5  Experimental details

We detail our experiments in dimension 1 in Appendix S5.1. In Appendix S5.2, we give details on our experiment on the synthetic mixture of two Gaussians derived from MNIST. Finally, we detail the experiment on the subset of all 3 and 7 of MNIST in Appendix S5.3. We trained our models using 2 NVIDIA Titan Xp from the proprietary server of our institution with an estimated total training time of approximately 175 GPU hours. Code is available here [3].

### S5.1  Univariate case

In the univariate case we use a simple 3-layer Multi Layer Perceptron (MLP) of shape $(1, 128, 256, 1)$ as decoder for the VAE and as generator for the GAN. The network has a total of 33537 learnable parameters. The score network uses also a a 3-layer MLP block, this time of shape $(1, 96, 196, 1)$, in which at each layer is injected the noise information transformed by a positional encoding (Vaswani et al., 2017) and then by another MLP block size $(16, 32, 64)$, see Figure S2. The score network has a total of 34665 learnable parameters. In all three models, we use LeakyReLU (Maas et al., 2013) as

---

[3]https://github.com/AntoineSalmona/Push-forward-Generative-Models

non-linearity with a negative slope of $0.2$. The three models are trained during $400$ epochs with a batch size of $1000$ using ADAM (Kingma and Ba, 2015) with a momentum of $0.9$ and a learning rate of $10^{-4}$. In the following, we give more specific details for each model.

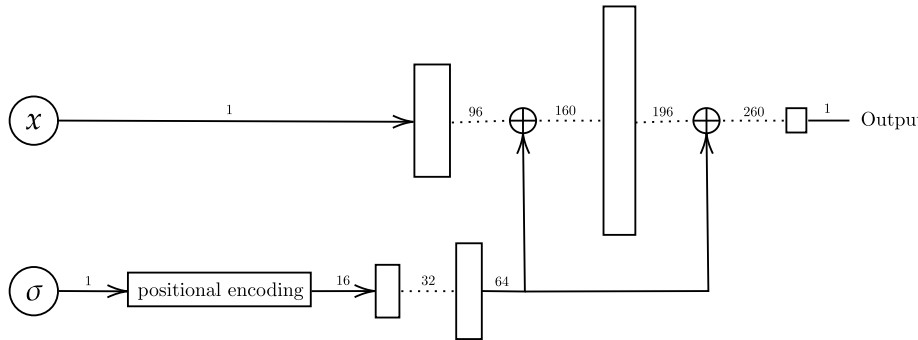

Figure S2: Architecture of the score network used for the univariate experiments. The "positional encoding" block applies the sine transform described in Vaswani et al. (2017). $\bigoplus$ corresponds to concatenation, the vertical blocks correspond to the fully connected layers and the numbers over the arrows correspond to the size of the vectors.

**Variational autoencoder.** We use the vanilla VAE model as described in Kingma and Welling (2014). In the following, we denote $\theta$ and $\phi$ the respective parameters of the decoder and the encoder. The decoder $f_\phi$ is composed of an MLP block of size $(1, 256, 128)$ followed by two parallel fully connected layers of shape $(128, 1)$ which gives two outputs $f_{1\phi}(x)$ and $f_{2\phi}(x)$. Then the input $z$ of the decoder $g_\theta$ is obtained by the so-called reparametrization trick, which consists in sampling $z \sim q_\phi^{z|x}$, where $q_\phi^{z|x} = \mathrm{N}(f_{1\phi}(x), \exp[f_{2\phi}(x)])$. During training, the model minimizes the following loss function:

$$\mathcal{L}_{\mathrm{VAE}}(\theta, \phi) = \mathbb{E}_{x\sim\nu}[\mathrm{ELBO}_{\theta,\phi}(x, q_\phi^{z|x}, p_\theta^{x|z})] \,,$$

where $p_\theta^{x|z} = \mathrm{N}(g_\theta(z), c^2\,\mathrm{Id}_d)$ and ELBO is the Evidence Lower Bound (Blei et al., 2017), defined as follows:

$$\mathrm{ELBO}_{\theta,\phi}(x, q_\phi^{z|x}, p_\theta^{x|z}) = \mathbb{E}_{z\sim q_\phi^{z|x}}[\log(p_\theta(x|z))] - d_{\mathrm{KL}}(q_\phi^{z|x}||\mathrm{N}(0, \mathrm{Id}_p)) \,.$$

The standard deviation $c$ in $p_\theta^{x|z}$ is an hyperparameter of the model. For our experiments, we observed that $c = 0.1$ gave good results.

**Generative adversarial network.** As for the VAE, we use the vanilla GAN model as described in Goodfellow et al. (2014). The discriminator is $4$-layer MLP of shape $(1, 512, 256, 128, 1)$ with spectral normalization (Miyato et al., 2018) in order to reduce as much as possible mode collapse. We train the model using the vanilla adversarial loss, that the discriminator $d_\phi$ tries to maximize and that the generator $g_\theta$ tries to minimize

$$\mathcal{L}_{\mathrm{GAN}}(\theta, \phi) = \mathbb{E}_{x\sim\nu}[\log(d_\phi(x)] + \mathbb{E}_{z\sim\mathrm{N}(0,\mathrm{Id}_p)}[\log(1 - d_\phi(g_\theta(z)))] \,.$$

We also tried with the hinge version of the adversarial loss, as proposed in Lim and Ye (2017) and Tran et al. (2017) and we obtained similar results.

**Score-based generative modeling.** Our diffusion model is similar to the model introduced by Song and Ermon (2019). The neural network $s_\theta$ learns to approximate, for a given $x$ and a given $\sigma$, the score $\nabla_x p_\nu(x, \sigma)$ of the data distribution convoluted with a Gaussian distribution of standard deviation $\sigma$. This is done by first defining a geometrical progression $\{\sigma_i\}_{i=1}^L$ where $L = 10$ and where the ratio is chosen such that $\sigma_L \approx 0.01$, and then minimizing the Fischer divergence (Vincent, 2011)

$$\mathcal{L}_{\mathrm{SGM}}(\theta) = \mathbb{E}_{\sigma\sim 1/L\sum\delta_{\sigma_i}}\left[\sigma^2\mathbb{E}_{x\sim\nu}\left[\mathbb{E}_{y\sim\mathrm{N}(x,\sigma^2\,\mathrm{Id}_d)}\left[\left\|s_\theta(y, \sigma) + (y - x)/\sigma^2\right\|^2\right]\right]\right] \,.$$

Then, in order to generate data, we use an annealed Langevin dynamic scheme as defined in Song and Ermon (2019). In the Langevin dynamic, we set the step size to $2 \times 10^{-5}$ and the number of step for each value of $\sigma$ to $100$ as in Song and Ermon (2019).

**Influence of generator depth.** For this experiment, we increase the number of layers of the VAE decoder and the GAN generator from 2 to 6. At each new layer, we double the number of neurons at the previous layer. For instance, the generative network with 2 layers is thus an MLP of shape $(1, 128, 1)$ and the one with 6 layers is an MLP of shape $(1, 128, 256, 512, 1024, 2048, 1)$. Specifically to the GAN model, we also increase the number of layers in the discriminator in order to keep the dynamic between this latter and the generator balanced. As in the 3-layers case, the discriminator is one layer deeper than the generator. For instance, the discriminator associated to the generator with 2 layers is an MLP of shape $(1, 256, 128, 1)$.

**Influence of generator architecture.** For this experiment, we use a feed-forward MLP of shape $(1, 256, 256, 256, 1)$ as backbone. Then we add two additive pre-activation skip-connections of type "resnet" between the first and the second hidden layers and between the second and the third hidden layers. Finally, we replace the two previous additive skip-connections of type "resnet" by concatenation pre-activation skip-connections of type "densenet".

## S5.2 Synthetic mixture of Gaussians on MNIST

**Models details.** We adapt our three models to MNIST, changing mainly the networks architectures and making small modifications that we describe in what follows. We base the architecture of the GAN and the VAE on DCGAN (Radford et al., 2015), using the generator as decoder and the discriminator as encoder for our VAE. This is done by doubling the last layer of the discriminator in order that the VAE encoder has two outputs as in the univariate case. For the GAN model, we replaced the convolutional discriminator by a simple MLP of shape $(784, 512, 256, 128, 1)$ because the dynamic between the generator and the discriminator seemed unbalanced otherwise. We also update our GAN model using some features of SAGAN (Zhang et al., 2019): applying spectral normalization on the discriminator and using the unconditional hinge version of the adversarial loss (Lim and Ye, 2017; Tran et al., 2017):

$$\mathcal{L}_{\text{GAN}}^{d_\phi} = -\mathbb{E}_{x\sim\nu}[\min\{0, -1 + d_\phi(x)\}] - \mathbb{E}_{z\sim\text{N}(0,\text{Id}_p)}[\min\{0, -1 - d_\phi(g_{\theta(z)})\}] ,$$
$$\mathcal{L}_{\text{GAN}}^{g_\theta} = -\mathbb{E}_{z\sim\text{N}(0,\text{Id}_p)}[d_\phi(g_\theta(z))] .$$

Such loss function is equivalent to minimize the Kullback-Leibler divergence between the generated distribution and the data distribution. The VAE decoder and the GAN generator have 1713088 learnable parameters. For the score network architecture, we use the vanilla U-Net architecture (Ronneberger et al., 2015) in which we double the number of channels at each layer, we add group normalization (Wu and He, 2018) after each convolution and we replace the ReLU non-linearies by SiLU (Elfwing et al., 2018). As in the univariate case, we use positional encoding (Vaswani et al., 2017) followed by a MLP block of shape $(1, 16, 32)$ to incorporate the noise information at each layer. The score network has 1607392 learnable parameters. For inference, we use the same Langevin dynamic scheme as above with the same hyperparameters as in the univariate case. The three models are trained during 100 epochs with a batch size of 128 using ADAM with a momentum of 0.9 and a learning rate of $2 \times 10^{-4}$.

**Additional details.** The histograms of projection on the line passing through the mean of each Gaussians are obtained using 20000 generated samples. To assign a color to each bin of the histograms, we train a simple MLP of shape $(784, 1024, 50, 10)$ as classifier on MNIST. The classifier is trained during 10 epochs using again ADAM with a momentum of 0.9 and a learning rate of $2 \times 10^{-4}$ and reaches an accuracy of 0.98 on the test set.

## S5.3 Subset of MNIST

**Models details.** Since the dataset is more complex than before, we use bigger models. For the score network, we use the architecture defined in Ho et al. (2020), in which we set the number of channels to 64 instead to 128 and we remove the self attention layers (Wang et al., 2018) for computational resource purposes. The score network has 6072065 learnable parameters. Again, we use an annealed Langevin dynamic scheme for inference with the same hyperparameters as before. For the VAE and the GAN, we use the same architecture as before, using this time the convolutional discriminator of DCGAN, and quadrupling the number of channels at each layer. This is mainly done in order to scale the generator/decoder to the score network. Hence the VAE decoder/GAN generator has 7151104

learnable parameters. We train all three models during 600 epochs with a batch size of 128 using ADAM with a momentum of 0.9 and a learning rate of $2 \times 10^{-4}$.

**Additional details.** We use the deep Wasserstein embedding proposed by Courty et al. (2018) in order to visualize histograms of projection in the Wasserstein space. We use the exact same network architecture and the same training procedure that in Courty et al. (2018): first, one million pairs of digits of MNIST are chosen randomly, in which 700000 are kept for the training set, 200000 for the test set, and 100000 for the validation set. We normalize each image in order to consider it as a two-dimensional distribution and we compute the 1-Wasserstein distance for each pair. Then, we train an autoencoder in a supervised manner in a way that the images at output of the autoencoder are close to the images in input, and that the euclidean distance between two vectors in the latent space is close to the 1-Wasserstein distance between the two corresponding images of MNIST. As in Courty et al. (2018), the latent Wasserstein space is of dimension 50 and the autoencoder is trained during 100 epochs with a batch size of 100 and with an early stopping criterion. Again, we use ADAM with a momentum of 0.9 and a learning rate of $10^{-3}$. We use the same classifier as before to assign color to each bin of the histograms. Finally, the histograms of projection on the line passing through the deep Wasserstein barycenters of all 3 and 7 are obtained using 20000 generated samples.

## S6 Additional experimental results

In the following, we provide additional experimental results. First, we compare estimates of the bounds of Theorem 4, Corollary 6, and Theorem 7 to estimates of the total variation distance and the Kullback-Leibler divergence in the univariate case. Then we study the possible correlation between the size of the score network and the tendency of the score-based model to generate unbalanced modes. Finally, we provide additional visualizations of histograms of generated distributions for the univariate case and generated samples for the experiments on MNIST.

### S6.1 Bounds on TV distance and KL divergence in the univariate case

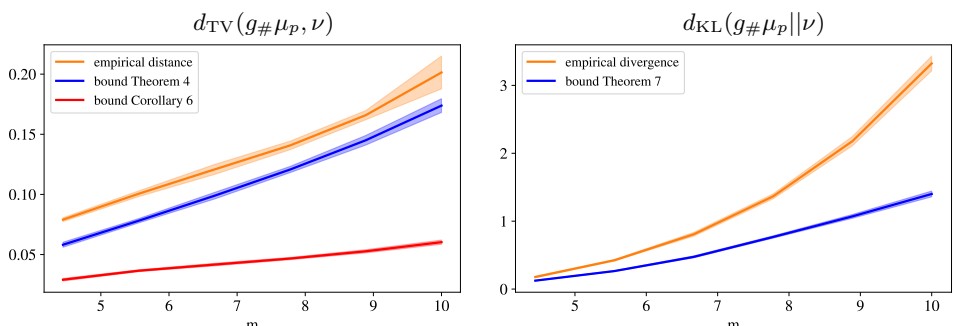

Figure S3: total variation distance (left) and Kullback-Leibler divergence (right) for the VAE (in orange) and estimates of the respective lower bounds from Theorem 4 and Theorem 7 in blue. The lower bound of Corollary 6 is also plotted in red for the total variation. The experiments are averaged over 10 runs and the colored bands correspond to +/- the standard deviation.

In this experiment, we compare estimates of the bounds of Theorem 4, Corollary 6, and Theorem 7 to estimates of the total variation distance and the Kullback-Leibler divergence. We only provide results for the VAE since the bounds are not interesting for the GAN since they are consequences of interpolation between modes due to a small Lipschitz constant of the generative network. Yet this latter in the GAN case achieves a large Lipschitz constant so does not interpolate significantly. To estimate empirically the total variation distance and the Kullback-Leibler divergence, we used their respective analytical formula

$$d_{\text{TV}}(g_{\#}\mu_p, \nu) = (1/2)\int_{\mathbb{R}} |p_{g_{\#}\mu_p}(x) - p_{\nu}(x)|dx,$$
$$d_{\text{KL}}(g_{\#}\mu_p||\nu) = \int_{\mathbb{R}} p_{g_{\#}\mu_p}(x) \log\left(p_{g_{\#}\mu_p}(x)/p_{\nu}(x)\right) dx,$$

where $p_{g_{\#}\mu_p}$ and $p_{\nu}$ are the respective densities of $g_{\#}\mu_p$ and $\nu$. In order to estimate the lower bounds of Theorem 4 and Theorem 7, we set A of the form $(-\infty, -r/2]$ and we perform a grid search on $r$.

In Figure S3, we can observe that the estimates of the bounds provided by Theorem 4 and Theorem 7 are not tights. This is possibly because we selected a sub-optimal A but it most likely follows from the fact that the bounds don't take into account that $g_{\#}\mu_p$ has automatically less mass on the modes than $\nu$ since a significant amount of its total mass is between them. One can also observe that the explicit lower bound of Corollary 6 is much smaller than the bound of Theorem 4. This can be explained by the facts that $\|m\|/2\sigma$ is probably a sub-optimal choice of $r$ and that the bound of Corollary 6 minimizes the interpolation between modes over all the mappings with Lipschitz constant $\mathrm{Lip}(g)$, regardless whether these mappings approximate well $\nu$ on its modes or not. Since there is less interpolation if the modes are unbalanced (see Section 3.2), it is likely that the mappings $g$ such that $g_{\#}\mu_p$ is unbalanced are affecting the value of this bound in a bad way.

## S6.2    Additional examples

### S6.2.1    Univariate histograms

We provide additional visualizations of histograms of generated data with the three models for various values of $m$.

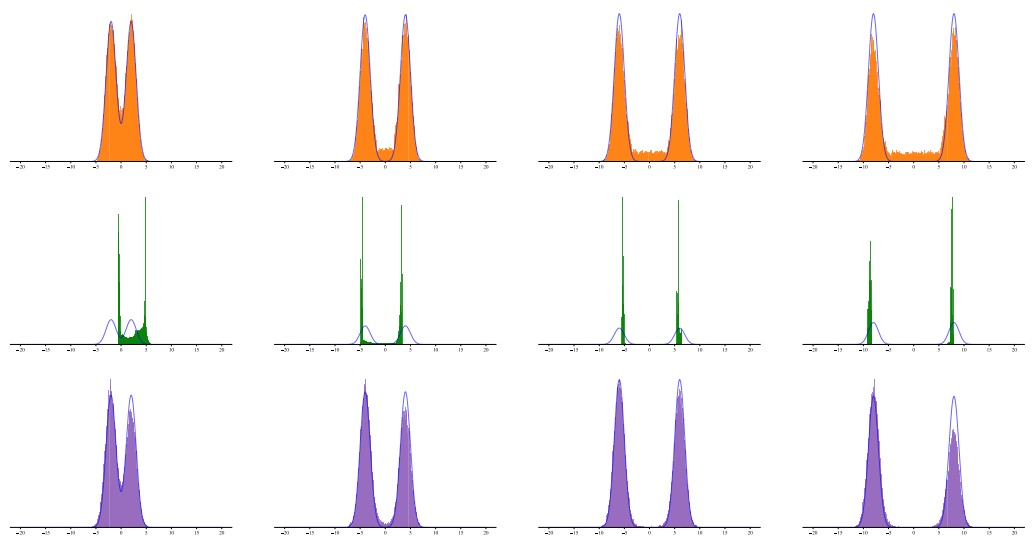

Figure S4: Histograms of distributions generated with VAE (top, in orange), GAN (middle, in green), and with SGM (bottom, in purple) for $m = 2$, $m = 4$, $m = 6$ and $m = 8$. The data distribution densities are plotted in blue.

We can observe that the score-based model already generates unbalanced modes, but the phenomenon is globally less visible than in higher dimensions. Secondly, we provide additional visualizations of histograms of generated data with GANs trained with an additional gradient penalty term in the generator loss for various values of $L \approx \mathrm{Lip}(g)$.

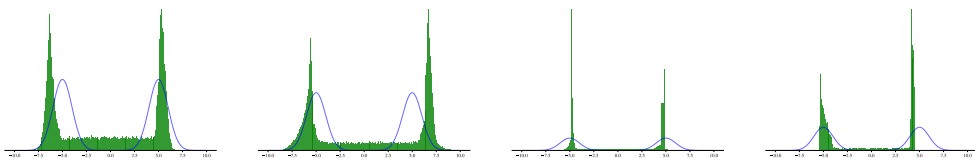

Figure S5: histograms of distributions generated with GANs with with gradient penalty for $\mathrm{Lip}(g) \approx L = 11$, $\mathrm{Lip}(g) \approx L = 15$, $\mathrm{Lip}(g) \approx L = 19$ and $\mathrm{Lip}(g) \approx L = 23$. The data distribution densities are plotted in blue.

### S6.2.2 Visualization of generated data

Finally, we show randomly chosen generated samples with VAE, GAN and SGM on the synthetic mixture of Gaussian on MNIST and the subset of all 3 and 7 of MNIST.

Figure S6: Generated samples with VAE, GAN and SGM on the synthetic mixture of Gaussian on MNIST (top) and the subset of all 3 and 7 of MNIST (bottom). The samples have been randomly chosen.