# OpenReview forum: "Can Push-forward Generative Models Fit Multimodal Distributions?"
_NeurIPS.cc/2022/Conference — NeurIPS 2022 Accept_

### Official Review · Reviewer_Wby3 · 2022-07-10

**Rating:** 5
**Confidence:** 4
**Soundness:** 3 good
**Presentation:** 3 good
**Contribution:** 2 fair

**Summary:**

The paper derives a lower bound of the Lipschitz constant of the push-forward function between the standard Gaussian distribution and the data measure. Or, in a complementary way, it provides a bound on the "distance" between the model distribution (defined by the standard Gaussian prior and the push-forward function) and data measure.

The paper further considers a mixture of two Gaussians to calculate the bound explicitly and vadiliate the theorom.

The paper evaluates VAE, GAN and SBM on mixtures of two Gaussians to explain certain known behavoiur of such models, e.g., mode collapse and so on.

**Questions:**

See weakness.

**Limitations:**

The limitations of the paper mentioned in weakness should be discussed.

This is a theory paper and I did not see a direct negative societal impact.


**Strengths And Weaknesses:**

I'm not that excited about the paper and I have the following concerns.

1. Generally, the results of Theorem 1 are not surprising. As far as I know, the role of Lipschitz constant in the expressiveness of deep generative models has been extensively studied, e.g. [1*] and many others.

2. The implications of the results are not new. Theorem 2-7 are either hard to evaluate or limited to the oversimplified mixture of two Gaussians. I did not see clear guidance on how to design a push-forward generator given practical assumptions of data. Besides, the behavior explained by this paper has been widely studied and explained from different perspectives. I did not see a particularly interesting point or obtain new insights from the theory.

3. According to my understanding, VAE uses a stochastic decoder and I'm not sure whether it can be formulated as a push-forward function.

4. I'm a little bit confusing about evaluating score-based models as well. Do such models fit the theory? The excellent empirical results of such models on a mixture of two Gaussians are not surprising given the fact that they are proven effective on much more complex real data. I don't think the theory of the paper explains the empirical results either.

[1*]  implicit normalizing flow.

---

> ### Author Response · Authors · 2022-08-02
> **Response to Reviewer Wby3 (END)**
>
> >I'm a little bit confusing on evaluating score based models as well. Do such models fit the theory? The excellent empirical results of such models on mixture of two Gaussians are not surprizing given the fact that they are proven effective on much more complex real data. I don't think the theory of the paper explains the empirical results either.
>
> As mentioned in the global response, SGMs can also be seen as push-forwards of Gaussian and so also fit the theory. The major difference with push-forward models (as GANs or VAEs) is that optimization is not directly performed on the push-forward mapping itself but on an auxiliary function. This is clarified in the revised version of our paper. Their good results on real complex data did not guarantee that they approximate well the data distribution since observing the data distribution is difficult in higher dimensions.

---

> > ### Comment · Reviewer_Wby3 · 2022-08-03
> > **Further questions**
> >
> > Thanks for the feedback. I have further questions based on the author's feedback to me and the global response.
> >
> > 1. Do Bernoulli-VAE and the full covariance VAE fit the push-forward formulation? It would be better if the authors explicitly discuss all types of models that fit the formulation of the paper. I mean explicitly stating the scope of the theory and discussing the ones that don't fit.
> >
> > 2. I'm still confused about the discussion of SGM in this paper. In particular, according to the general response, the authors take the empirical results of SGM (as well as a guessed reason) as one of the two major contributions of the paper. I think this is a theoretical paper while the empirical behavior of SGM is not formally explained by the theory. Have you ever tried to quantitatively show that the Lipstchiz lower bound by the score representation is much larger than the push-forward mapping (assuming a similar compacity of the neural network including architectures and weight norm)?  Besides, I think SGM can be understood as a very deep hierarchical VAE (e.g., 1000 stochastic latent layers) during generation, then its Lipstchiz constant is roughly 1000 times larger than VAE. Does this directly explain the behavior of SGM?
> >
> > Currently, mixing the theory and SGM results together in the paper is misleading and hurts the organization of the paper.

---

> > > ### Author Response · Authors · 2022-08-04
> > > **Response to Reviewer Wby3**
> > >
> > > Thank you for your feedback. We submitted a new version of the paper and we modified the general response to correct some of the issues you address.
> > > ***
> > > >Do Bernoulli-VAE and the full covariance VAE fit the push-forward formulation? It would be better if the authors explicitly discuss all types of models that fit the formulation of the paper. I mean explicitly stating the scope of the theory and discussing the ones that don't fit.
> > >
> > > - Bernoulli-VAEs are not designed for real-valued data, and so are not included in the scope of this paper.
> > > - The only difference between Full covariance VAEs and standard Gaussian-VAEs is in the reparametrization trick at the end of the encoder. The generation process in both models is the same.
> > > - We added a footnote which states that the VAE model considered in this paper is the Gaussian-VAE. Upon acceptance of the paper, we will add such a discussion on which models fit the theory or not in the supplementary material.
> > >
> > > N.B: it seems that in our previous response, we misused the term “posterior”: the posterior probability is $ p(z|x) $ and not $ p(x|z) $. Sorry for this confusion.
> > > ***
> > > >I'm still confused about the discussion of SGM in this paper. In particular, according to the general response, the authors take the empirical results of SGM (as well as a guessed reason) as one of the two major contributions of the paper. I think this is a theoretical paper while the empirical behavior of SGM is not formally explained by the theory.
> > >
> > > - It seems that the formulation “our contribution is twofold” in the Global Response was confusing. We wanted to emphasize the point on SGMs, but we do not claim this is one of the two major contributions of the paper. Our contributions remain obviously mainly theoretical. We modified the Global response accordingly to avoid any confusion in the future.
> > > - We have also modified the article to better distinguish what we actually show on SGMs and what is our interpretation.
> > > - We agree that the theory doesn’t formally explain the empirical behavior of SGMs. The theory gives insights on why it is difficult to fit multimodal distributions with VAEs and GANs and why it is easier with SGMs.
> > > ***
> > > >Have you ever tried to quantitatively show that the Lipstchiz lower bound by the score representation is much larger than the push-forward mapping (assuming a similar compacity of the neural network including architectures and weight norm)?
> > >
> > > - It is difficult to formally demonstrate that the Lipschitz constant of the whole generation dynamic is much larger than the score network principally because such a theoretical results would imply to find a lower bound on the Lipschitz constant of the generation dynamic which can roughly be seen as a composition of Lipschitz mapping. Yet, if deriving upper bounds on the Lipschitz constants of compositions of mapping is easy, deriving lower bounds is more difficult.
> > > - It is also difficult to quantitatively measure the empirical Lipschitz constant of the whole generation dynamic since this latter is a mapping from $ \mathbb{R}^{d(N+1)} $ ($ N $ being the total number of step in the dynamic) to $ \mathbb{R}^{d} $, and so $ d(N+1) $ is necessarily large.
> > > ***
> > > >Besides, I think SGM can be understood as a very deep hierarchical VAE (e.g., 1000 stochastic latent layers) during generation, then its Lipstchiz constant is roughly 1000 times larger than VAE. Does this directly explain the behavior of SGM?
> > >
> > > This is globally the intuition we have, that the structure of the generation dynamic consisting of stacked networks with noise inputs at each step allows to model easily mappings with large Lipschitz constants, which could explain partly the behavior of SGMs.

---

> > > > ### Comment · Reviewer_Wby3 · 2022-08-04
> > > > **Thanks.**
> > > >
> > > > Hi, thanks for the further clarification, which is quite helpful. My concerns 3&4 are addressed and I increased my score.

---

> ### Author Response · Authors · 2022-08-02
> **Response to Reviewer Wby3**
>
> Thank you for your review and for rising your concerns.
> ***
> >Generally, the results of Theorem 1 are not surprizing. As far as I know, the role of Lipschitz constant in the expressiveness of deep generative models has been extensively studied, e.g. [1*] and many others.
>
> - We agree that the results of Theorem 1 are intuitive and have been already observed in the Litterature, as for instance in [1*]. However, to the best of our knowledge, it hadn’t been formally demonstrated before. As Reviewer K1r7  said: “While the main claim of this paper is intuitive and somewhat expected, it is always good to formalize the intuition”. Theorem 1 shows that it is impossible to construct a generator with a small Lipschitz constant which pushes a Gaussian into a complex multimodal distribution, even if it was unlikely that such a generator existed. We agree that we didn't clarify enough what were the actual contributions of the paper in the previous version. This has been clarified in the revised version of the paper.
> - More specifically, in [1*] it is first said in the introduction "However, the Lipschitz constant of each transformation block is constrained for invertibility. In general, this is not preferable because mapping a simple prior distribution to a potentially complex data distribution may require a transformation with a very large Lipschitz constant". Then it is shown that relaxing the Lipschitz constraint of ResFlows allows to modelize a larger family of functions while using less blocks (which is a different result of the ones of this paper).
> ***
> >The implications of the results are not new. Theorem 2-7 are either hard to evaluate or limited to the oversimplified mixture of two Gaussians. I did not see clear guidance on how to design a push-forward generator given practical assumptions of data. Besides, the behaviour explained by this paper have been widely studied and explained in different perspectives. I did not see a particularly interesting point or obtain new insights from the theory.
>
> The motivation of this paper is not to improve GANs or VAEs performance. We agree that most of the implications of the theory for GANs and VAEs have been already observed in the literature since our results are intuitive. However, we think that the following aspects are new:
> - Some details given by the theory hadn't been mentioned to the best of our knowledge, as for instance the dimension-free feature of the problem or the influence of the modes proportion.
> - Score-based generative models (SGM) are able to circumvent the limitation that the theory highlights, by not optimizing directly
> the push-forward mapping itself, but an auxiliary function (the score).
> - The comparative aspect of the different models, which links different behaviors observed in different contexts.
> ***
> >According to my understanding, VAE uses a stochastic decoder and I'm not sure whether it can be formulated as a push-forward function.
>
> In the original formulation of VAE [2*], the conditional probability (when the data a real valued) is modeled by $ p(x|z) = \mathcal{N}(g_{\theta}(z),h_{\theta}(z)^2I_d) $, where $ (g_{\theta}(z),h_{\theta}(z)) $ are the outputs of a deterministic neural network (the decoder). In practice, this model is most of the time simplified to $ p(x|z) = \mathcal{N}(g_{\theta}(z),c^2I_d) $, where $ g_{\theta}(z) $ is the single output of the decoder and c > 0 is an hyperparameter of the model.
> - If relaxing the conditional probability to $ p(x|z) = \mathcal{N}(g_{\theta}(z),c^2I_d) $ or $ p(x|z) = \mathcal{N}(g_{\theta}(z),h_{\theta}(z)^2I_d) $ instead of $ p(x|z) = \delta_{g_{\theta}(z)} $ is crucial for the training of VAEs, this is not desired during the generation process since $ \mathcal{N}(g_{\theta}(z),c^2I_d) $  or $  \mathcal{N}(g_{\theta}(z),h_{\theta}(z)^2I_d) $  are just noisy versions of the push-forward measure  (sampling from the relaxed conditional corresponds to sample from the Gaussian measure pushed by $ g_{\theta}$  and then to add isotropic noise).
> - Even if we sample from the relaxed conditional probability during generation, this is still a push-forward generative model
> since the generated data are of the form $ G_{\theta}(z,z') $ where $ z $ and $ z'$ are Gaussian vectors on respectively the latent space and the ambient space, and  $ G_{\theta} $ is the deterministic neural network defined by $ G_{\theta}(z,z') = g_{\theta}(z) + cz' $ if $ p(x|z) =  \mathcal{N}(g_{\theta}(z),c^2I_d) $ and $ G_{\theta}(z,z') = g_{\theta}(z) + h_{\theta}(z)z' $ if $ p(x|z) =  \mathcal{N}(g_{\theta}(z),h_{\theta}(z)^2I_d) $.
> ***
> [1*] Implicit Normalizing Flow
> [2*] Auto-Encoding Variational Bayes

---

### Official Review · Reviewer_k9qb · 2022-07-11

**Rating:** 6
**Confidence:** 3
**Soundness:** 3 good
**Presentation:** 3 good
**Contribution:** 3 good

**Summary:**

This paper derives a lower bound for the total deviation distance, and KL-divergence between the distribution learnt by push-forward generative models (VAEs-GANs), and the true data distribution. According to their lower bound (corollary 6), the lower bound would be larger for more multi-modal distributions, but the Lipschitz constant is also a determinining factor, in that a larger Lipschitz constant should be able to compensate for the large degree of multi-modality.

The authors also concretize their theoretical findings by providing three main experiments.

1 - They train GAN, VAE, Score-based generative model on 1D-Mixture of Gaussian data. They find that VAEs consistently generate significant amount of data in the region in-between modes, due to its inflexibility reflected by its small Lipschitz constant. GANs are more flexible with a larger Lipschitz constant, but they fail to capture the whole distribution. Score-based models however better capture the distribution, even though they have smaller Lipschitz constant than GANs as shown in Figure 2.

2 - They train the same three models, on a 784 dimensional-GMM data where means are located on two specific MNIST-digits, from the classes 3 and 7. We see that the behavior is pretty much the same as the 1D GMM data, except the score based model is not able to learn the correct proportion for each mode this time.

3 - They train the same three models on the all 3-7 digits from the MNIST dataset. We now actually see that the data is less multi-modal, and therefore VAEs and GANs work better.



**Questions:**

As you pointed out in the discussion section, there also exists techniques that use learnable / multimodal prior distributions. And this significantly alleviates the burden on the decoder / encoder. Given this, I wonder how relevant is your lower bound for generative models that push a unimodal prior distribution through the decoder. Do you think the theory that you provide in this paper could be extended to models with multimodal priors (e.g. GMM priors)?

I haven't checked all the details of the provided theorems and corollaries, but unlike corollary 6 where you provide a lower bound total variation distance, in theorem 7 the result does not seem to be final. Meaning, you still have the Borel sets in the result, and so it is not immediately clear to me what does that imply in practice. Do you think it would be possible to provide a result similar to corollary 6, in theorem 7 as well? Even considering a special case where the data distribution is Gaussian could be an improvement in my opinion.



**Limitations:**

I think the main limitation of the theory provided in this paper is due to that fact that it does not consider multi-modal priors. I think that multi-modal priors are crucial to mitigate the stiffness of VAEs. If however, the theory in this paper could be extended for multi-modal priors, it could be a nice first step.


**Strengths And Weaknesses:**

Strengths:

This paper provides potentially useful insights on the relationship between the level of multimodality and the expressivity of the push-forward generative models.

I think that the numerical experiments really help to consolidate the message regarding multimodality and the behavior of the three main types of generative model learning methods.


Weaknesses:

For a non-theory researcher working with generative models, I think that this paper could be useful to read. However, I also think that section 3 could be improved for reader-friendliness. My suggestion would be to clearly highlight the main results. (Corollary 6 and Theorem 7, right?) It might be useful to highlight the main result (the actual lower bound as opposed to alluding to it with text) in the introduction actually.

---

> ### Author Response · Authors · 2022-08-02
> **Response to Reviewer k9qb**
>
> Thank you for your positive review and your suggestions. We glad that you found interest in reading this paper.
> ***
> >For a non-theory researcher working with generative models, I think that this paper could be useful to read. However, I also think that section 3 could be improved for reader-friendliness. My suggestion would be to clearly highlight the main results. (Corollary 6 and Theorem 7, right?) It might be useful to highlight the main result (the actual lower bound as opposed to alluding to it with text) in the introduction actually.
>
> The main theoretical results of this paper are actually Theorem 1, Theorem 4, and Theorem 7 which formalize the intuitive fact that the Lipschitz constants of the push-forward maps has to be large in order to approach correctly a given multimodal distribution.  The corollaries can be seen as simple toy-examples which illustrate the abstract theorems. We agree that the paragraph "Contributions of the paper" at the end of the introduction was misleading in the previous version of the paper and so this has been clarified in the revised version.
> ***
> >As you pointed out in the discussion section, there also exists techniques that use learnable / multimodal prior distributions. And this significantly alleviates the burden on the decoder / encoder. Given this, I wonder how relevant is your lower bound for generative models that push a unimodal prior distribution through the decoder. Do you think the theory that you provide in this paper could be extended to models with multimodal priors (e.g. GMM priors)?
>
> We expect that our results can be extended to multimodal priors. In particular, we expect that the push-forward models will present the same limitation as in the unimodal prior case if the number of modes in the latent space is smaller than the number of modes in the target space. To extend the theory, we would probably need new tools in this case because multimodal priors do not satisfy the Gaussian isoperimetric inequality.
> ***
> >I haven't checked all the details of the provided theorems and corollaries, but unlike corollary 6 where you provide a lower bound total variation distance, in theorem 7 the result does not seem to be final. Meaning, you still have the Borel sets in the result, and so it is not immediately clear to me what does that imply in practice. Do you think it would be possible to provide a result similar to corollary 6, in theorem 7 as well? Even considering a special case where the data distribution is Gaussian could be an improvement in my opinion.
>
> We added a corollary of Theorem 7 when the data distribution is a mixture of two isotropic Gaussians in the supplementary material. The lower bound is in that case:
> $$ d_{KL} \geq  A\log\left(\frac{A}{B}\right) + (1-A)\log\left(\frac{1-A}{1-B}\right),$$
> with
> $$ A = \int_{-\Phi^{-1}(1-\lambda)}^{||m_2 - m_1||/4\sigma\text{Lip}(g) -\Phi^{-1}(1-\lambda)} \varphi(t)dt \ \text{and} \ B = (1/2)\int_{||m_2 - m_1||(2\sigma - 1)/4\sigma^2}^{||m_2 - m_1||(2\sigma  + 1)/4\sigma^2}\varphi(t) dt, $$
> where $ \lambda \in (0,1/2] $ depends on $ g $ and corresponds to the proportion of the modes in the generated distribution. Note that this time, the Lipschitz constant is not the only dependence in $ g $.

---

### Official Review · Reviewer_K1r7 · 2022-07-11

**Rating:** 7
**Confidence:** 4
**Soundness:** 3 good
**Presentation:** 3 good
**Contribution:** 3 good

**Summary:**

This paper focuses on the expressivity of "push-forward" generative models trained on multimodal distributions. Specifically, a series of bounds are derived to relate the Lipschitz constant of a generative model to the total variation distance and the KL divergence between the generated and target distributions. Empirical analyses based on synthetic data and MNIST confirm the theoretical results.

**Questions:**

It would be interesting to see experiments on more realistic datasets, as well as on more diverse generative models, such as energy-based models.

**Limitations:**

The authors have addressed a potential limitation in their theoretical analysis.

**Strengths And Weaknesses:**

Strengths

While the main claim of this paper is intuitive and somewhat expected, it is always good to formalize the intuition. Moreover, the theoretical analyses are thorough and well-presented, providing bounds for both the Lipschitz constant and the probability distance measures.

Weaknesses

It seems the limited expressivity of push-forward generative models results from using simple feedforward NNs for generators, rather than from the VAE and GAN objectives. This distinction should be further discussed, and could be verified experimentally by using more expressive generator architectures.

The empirical study only uses simple synthetic data and the MNIST dataset, thus it is not clear how well the results will generalize to more realistic datasets.

---

> ### Author Response · Authors · 2022-08-02
> **Response to Reviewer K1r7**
>
> Thank you for your positive feedback and your suggestions.
> ***
> >It seems the limited expressivity of push-forward generative models results from using simple feedforward NNs for generators, rather than from the VAE and GAN objectives. This distinction should be further discussed, and could be verified experimentally by using more expressive generator architectures.
>
> We agree this is an important question that we did not considered in the previous version of the paper. We have added an experiment (in the supplementary material, but we plan to add it in the main in the future version of the paper) where we compare three different architectures of the generative network for the VAE and the GAN in the univariate case:
> - A simple MLP of shape (1,256,256,1) that we use as backbone for the two other architectures.
> - The backbone with additive skip-connections of type "resnet" [1*].
> - The backbone with concatenation skip-connections of type "densenet" [2*].
>
> This experiment gives mainly two insights:
> - In the VAE setting, using more advanced decoder architectures doesn't seem to help reaching larger Lipschitz constants. This supports
> the idea that the Lipschitz constant of the decoder is naturally limited by the VAE objective, as it has been indirectly demonstrated in [3*],
> and so the expressivity of VAEs is thus also limited.
> - In the GAN setting, the problem is not the expressivity in itself since GANs have shown to be able to reach large Lipschitz constants, but rather the stability.  To that extent, the resnet model seems to be more stable than the two others. This suggests that some architectures of the generative network might be better than others for learning mappings with large Lipschitz constants while staying stable.
> ***
> >The empirical study only uses simple synthetic data and the MNIST dataset, thus it is not clear how well the results will generalize to more realistic datasets. It would be interesting to see experiments on more realistic datasets, as well as on more diverse generative models, such as energy-based models.
>
> We agree that an experiment on more realistic datasets and on more diverse generative models would be an improvement for the paper. However, we are limited by our computational resources  (4 NVIDIA titan XP) and we cannot run experiments on real images before the deadline, but we would be happy to run such experiments for a future version of the paper.
> ***
> [1*] Deep Residual Learning for Image Recognition
> [2*] Densely Connected Convolutional Networks
> [3*] On Implicit Regularization in β-VAEs

---

### Official Review · Reviewer_JkRJ · 2022-07-22

**Rating:** 5
**Confidence:** 3
**Soundness:** 2 fair
**Presentation:** 3 good
**Contribution:** 2 fair

**Summary:**

This paper studies the expressivity and stability of generative model that transforming a gaussian to data (such as GAN and VAE). In particular, the authors showed that there is a tradeoff between the approximation power and the stability (in terms of Lipchitz constant) of training for multimodal distribution. Experiment results are provided to validate the theoretical results.

**Questions:**

I would appreciate if the following questions could be answered:
1. In line 100 it is mentioned that in most of the time ‘p is almost always smaller than d in practice’. However, for corollary 2 and 3, in line 135 it is mentioned that p >= d. I was wondering if authors could comment on this.
2. Influences of model size and time of training paragraph in the experiment section: in line 277, I was wondering why ‘depth and the training time allow for greater expressivity’. It seems to me that only the Lipchitz constant is given in Figure 4, which I believe can be thought as a measure of the training stability. On the other hand, I do not seem to see which quantity measures the expressivity. In particular, I am not sure why longer training time could allow for better expressivity since it does not change the architecture of the network.


**Limitations:**

The limitations of the work are clearly discussed. This is a theoretical work thus I don’t see any potential negative social impact.

**Strengths And Weaknesses:**

Originality
1. The lower bound on Lipchitz constant (Theorem 1) and TV/KL distance (Theorem 4/7) seems to be new.

Quality
1. The overall quality of this paper is good. Theoretical results are provided with proof sketch. Detailed proofs are provided in the appendix. Limitations are clearly discussed.

Clarity
1. The paper is overall well-written and easy-to-follow.

Significance
1. Studying the expressivity and stability of push forward models is an important problem.
2. These theoretical results suggest a fundamental limitation of ‘push-forward’ generative models like GAN and VAE to achieve both good expressivity and stability (Lipschitz constant) for multimodal distribution. In particular, it shows that if the Lipchitz constant is small, then the model cannot recover the target distribution.
3. Recently scored-based generative models/diffusion models are becoming popular. Though the paper does not study these models in the theoretical analysis, some experiments on simple datasets are provided to show the differences between diffusion models and ‘push-forward’ models.

Minor
1. Line 33, dimension d -> dimension p?

---

> ### Author Response · Authors · 2022-08-02
> **Response to Reviewer JkRJ**
>
> Thank you for your review and your comments.
> ***
> >In line 100 it is mentioned that in most of the time ‘p is almost always smaller than d in practice’. However, for corollary 2 and 3, in line 135 it is mentioned that p >= d. I was wondering if authors could comment on this.
>
> - In real applications, as for instance image generation, the data distribution lives on a manifold of much lower dimension than the dimension of he ambient space.  This is the main reason why $ p $ is always smaller than $ d $ in GANs and VAEs in practice. However, in the setup of Corollary 2 and 3, the support of the target distribution is the whole ambient space itself. Since the image of $ \mathbb{R}^p $ through $ g $ is a manifold of dimension at most $ p $, it implies that $ p $ must be larger than $ d $ in that case in order to approximate correctly the target distribution.
> - It is important to note that the fact that $ p $ must be larger than $ d $ in the toy example of the mixture of Gaussian (whereas $ p$ is most of the time smaller than $ d $ for GANs and VAEs trained and real data) doesn't follow from a limitation of the theory but is rather a choice of
> presenting simple toy-examples in order to illustrate the abstract results of Theorem 1. Indeed, Theorem 1 is dimension free in the sense that nor $ p $ nor $d $ intervene directly in the bounds. Observe that $ p $ and $ d $ do not intervene either in the bound of Corollary 2. As a consequence, one could expect to find the same bound if we consider degenerate mixtures of Gaussians lying on lower dimensional manifolds than the ambient space (for instance a mixture on a line on $ \mathbb{R}^2 $) by applying the same reasoning that for Corollary 2, but intrinsically to the manifold.
> - Finally, we modified the original phrase of the paper by
> "Note that the support of $ \gamma $ and $ A $  can be sets of intrinsic dimension smaller than $ d $, which is most of the time the case when working with real data which are likely to live on low dimensional manifolds"
> since we feel this is less confusing and because in the setup of score-based models, we have automatically $ p > d $.
> ***
> >Influences of model size and time of training paragraph in the experiment section: in line 277, I was wondering why ‘depth and the training time allow for greater expressivity’. It seems to me that only the Lipschitz constant is given in Figure 4, which I believe can be thought as a measure of the training stability. On the other hand, I do not seem to see which quantity measures the expressivity. In particular, I am not sure why longer training time could allow for better expressivity since it does not change the architecture of the network.
>
> We agree that this phrase is wrong since the method doesn’t change when increasing the numbers of epochs and so longer training time fundamentally can't improve expressivity. Thank you for pointing this out, this has been modified in the revised version of the paper. We simply wanted to say that in the GAN setting, longer trainings allow to obtain generators with larger Lipschitz constants, but at the cost of stability. Related to this, we added an experiment which studies the impact of the architecture of the generative network (and thus the expressivity of the model) on its Lipschitz constant and on the training stability in the supplementary material (we plan to add it to the main in the future version of the paper).

---

### Official Review · Reviewer_tYcL · 2022-07-24

**Rating:** 5
**Confidence:** 4
**Soundness:** 4 excellent
**Presentation:** 4 excellent
**Contribution:** 3 good

**Summary:**

The paper studies Lipschitz lower bound of push-forward mappings on expression multi-modal target distributions. A particular focus is put on two-modal mixture distributions. The major discovery is the connection between how well a push-forward mapping represents a target mixture distribution and the mapping's Lipschitz continuity. Besides push-forward distribution generation, in experiments, score-based models are tested and compared with GANs and VAEs; experimental results support theory.

**Questions:**

Theories in the paper appear to be sound and correct. The following questions aim for a better understanding of interpretations and implications.

1. The dimension of latent distribution does not directly interact with the Lipschitz continuity of push-forward maps. Is it true that multi-dimensional latent distribution does not help in generating high-dimensional distributions? As Corollary 3 suggested, the continuity of any push-forward mapping cannot be better than the transport plan on transforming a univariate distribution. This seems discouraging.

2. Is the lower bound in Theorem 1 tight?

3. Can we discuss the scale of total variation lower bound and KL lower bound? For example, in Corollary 5, if the distance between the two manifold is $o(1)$, then the total variation lower bound is almost trivial, as the right-hand side is approximately 0, as long as ${\rm Lip}(g) = O(1)$. On the other hand, when dimension $d$ is large, the distance function can easily be very large, in the order of $O(\sqrt{d})$. This suggests the Lipschitz constant of $g$ should be at least $\sqrt{d}$ for a successful distribution generation.

Some additional questions for the experiments:

1. Is there any trend that increasing the Lipschitz continuity gives rise to better distribution generation? Figure 3 demonstrates a comparison. However, it is hard to tell whether the generated distribution becomes better, especially for $L = 15$ and $L = 25$. We see a clear trend of capturing modes though.

2. I feel like the experiments on score-based models does not help to support and clarify the theory, if I am not misunderstanding.

**Limitations:**

Authors already mentioned the tightness of total variation and KL lower bounds. I am mainly concerned with practical implications.

1. There is no clear message of how the Lipschitz continuity lower bound helps to improve performance of GANs in practice.

2. Theories focus on a relatively restrictive setting. In practice, different modes can overlap and there are often more than 2 modes.

**Strengths And Weaknesses:**

The paper is well written with few minor typos. The theoretical results are interesting and to the best of my knowledge, are new, at least newly applied for analyzing push-forward distribution generation. I like the presentation of the results, beginning with a useful technical theorem and then applying it to different instances. Experimental results are neatly summarized and provide support to theory.

No outstanding weaknesses as one may perceive; I do have some questions in the following section. I will also comment on limitations in the limitation section.

---

> ### Author Response · Authors · 2022-08-02
> **Response to Reviewer tYcL**
>
> Thank you for your positive comments and your thoughtful questions.
> ***
> >The dimension of latent distribution does not directly interact with the Lipschitz continuity of push-forward maps. Is it true that multi-dimensional latent distribution does not help in generating high-dimensional distributions? As Corollary 3 suggested, the continuity of any push-forward mapping cannot be better than the transport plan on transforming a univariate distribution. This seems discouraging.
>
> According to our theoretical results, the dimension of the latent space has no influence on the value of the Lipschitz constant necessary to push a Gaussian into a given measure with a Lipschitz mapping.  Yet, since the image of $ \mathbb{R}^p$  through the Lipschitz map  $ g$ is a manifold of dimension at most $ p $, it should be noted that the dimension of the latent space should not be smaller than the intrinsic dimension of the manifold on which the target distribution lies in order to correctly approach it. Since this intrinsic dimension of data is not known in real applications, it is always good to choose a latent dimension not too small to evict this case even if it seems to not directly help.
>
> >Is the lower bound in Theorem 1 tight?
>
> The second lower bound in Theorem 1 seems to be tight as it is shown in Figure 2 right. For the first bound, it is not straightforward to show whether the bound is tight or not. Yet, In the particular univariate case, it is clear that the bound is tight since it is reached when $ g $ is the Monge map which transports optimally the Gaussian to the target distribution (Corollary 3).
>
> >Can we discuss the scale of total variation lower bound and KL lower bound? For example, in Corollary 5, if the distance between the two manifold is o(1), then the total variation lower bound is almost trivial, as the right-hand side is approximately 0, as long as Lip(g)=O(1). On the other hand, when dimension d is large, the distance function can easily be very large, in the order of O(d). This suggests the Lipschitz constant of g should be at least d for a successful distribution generation.
>
> You're right. Let's consider the balanced case where the two manifolds have the same weights. In this case $\Phi^{-1}(\lambda) = 0$ and the lower bound in Corollary 5 is small only if Lip(g) is huge in comparison to the distance between the two manifolds. If both values have the same order of magnitude, the integral is non trivial and positive. We will be happy to add such a discussion when we will be less limited in terms of pages.
>
> >Is there any trend that increasing the Lipschitz continuity gives rise to better distribution generation? Figure 3 demonstrates a comparison. However, it is hard to tell whether the generated distribution becomes better, especially for L=15  and L=25. We see a clear trend of capturing modes though.
>
> As the Lipschitz constant increases, the model becomes better at separating the two modes of the distribution but at the cost of training instabilities and mode collapse. In Figure 3, when $ L = 25$, both modes are better captured than when $ L = 15 $ but more parts of the support of the target distribution are ignored. It seems that increasing the Lipschitz continuity allows to be better in terms of mode separation but worse in terms of accuracy of mode generation.
>
> >I feel like the experiments on score-based models does not help to support and clarify the theory, if I am not misunderstanding.
>
> The experiments on SGMs aim to empirically show they are able to circumvent the problem that the theory highlights. As mentioned in the global response, SGMs can also be seen as push-forwards of Gaussian and so also fit the theory. The major difference with push-forward models (as GANs or VAEs) is that optimization is not directly performed on the push-forward mapping itself but on an auxiliary function. This is clarified in the revised version of our paper.
> ***
> >There is no clear message of how the Lipschitz continuity lower bound helps to improve performance of GANs in practice.
>
> The motivation of this paper is not to improve the performance of GANs but to explain why such models perform badly on multimodal datasets by formalizing the intuitive fact that when the target distribution is multimodal, the push-forward map must have a large Lipschitz constant and is hence difficult to learn. Yet we provide experiments with SGMs specifically to show that they do not suffer from the same limitation.
>
> >Theories focus on a relatively restrictive setting. In practice, different modes can overlap and there are often more than 2 modes.
>
> We have chosen to illustrate our results with simple bi-modal distributions with well separated modes mainly in order to help comprehension. Yet the bounds of Theorems 1, 4 and 7 still apply when the modes overlap and there are more than 2 modes. To emphasize that, we added a generalization of Corollary 5 with more than 2 manifolds in the supplementary material.

---

> > ### Comment · Reviewer_tYcL · 2022-08-09
> > **Thank you for your response.**
> >
> > Thanks for the detailed response to my questions and concerns.
> >
> > SGM tries to estimate the score function (score matching) and according to the theory should give rise to a model of large Lipschitz constant. Figure 2 left, however, seems to indicate a different story as the lipschitz coefficient of SGM is significantly smaller than GANs. Am I missing anything?
> >
> > Most of my other concerns are addressed in the response. I am willing to keep my initial score.

---

> > > ### Author Response · Authors · 2022-08-09
> > > **Response to the new question**
> > >
> > > Thank you for your response.
> > > ***
> > > > SGM tries to estimate the score function (score matching) and according to the theory should give rise to a model of large Lipschitz constant. Figure 2 left, however, seems to indicate a different story as the lipschitz coefficient of SGM is significantly smaller than GANs. Am I missing anything?
> > >
> > > - Figure 2 left shows the Lipschitz constant of the neural networks involved in the different models. Yet in the SGM setting, the Lipschitz constant of the score network is not the Lipschitz constant of the push-forward mapping itself (which is the whole generation dynamic).
> > > Thus, since the theory implies that the Lipschitz constant of the push-forward mapping must necessarily be large in order to generate correctly the Gaussian mixture, Figure 2 left suggests that SGMs can (indirectly) learn push-forward mappings with large Lipschitz constants while keeping the Lipschitz constant of the score network relatively small. Our interpretation of this behavior is that mainly follows from the fact that the score network is applied numerous time during the generation process.
> > > - We agree that it would have been clearer to show directly the Lipschitz constant of the whole generation dynamic rather than the Lipschitz constant of the score network. Yet, it is difficult to quantitatively assess the empirical Lipschitz constant of the whole generation dynamic since this latter is in that case a mapping from $ \mathbb{R}^{N+1} $ to $ \mathbb{R} $, where $ N$ is the total number of steps in the generation dynamic.

---

### Author Response · Authors · 2022-08-02
**Global response to all reviewers**

We thank the reviewers for all their comments and their very constructive feedback. In the light of the reviews, we feel that some aspects of the paper had to be clarified. We want to emphasize that the main motivation of this paper is not to give new insights in order to improve the performances of the different models but to mathematically understand the difficulties of generating multimodal distributions with deep generative models.
- Our main contribution is to formally demonstrate that the Lipschitz constant of generative networks must be large in order to push a simple Gaussian to a complex multimodal distribution. Although this result is intuitive and was already mentioned by several works in the literature, it had not been proven formally yet. We think it was particularly important to demonstrate such a result since intuition can be misleading when working with generative modeling in high dimensions. Since common practice for training GAN or VAE is to limit their Lipschitz constants to stabilize the training, the main practical implication of our contribution is that as such, these generative networks will always have difficulties generating complex distributions.
- We also wanted to show in our experiments that Score-based Generative Models (SGMs) do not meet the same limitations. Observe that in SGMs,  generated data are also of the form $ g(z) $ where  $ g $  is the whole reverse diffusion dynamic and $ z $ is a Gaussian vector of size $  d(N+1) $, $ N $ being the total number of steps in the generation dynamic. Thus these models can also be seen as
push-forwards of Gaussian distributions and so fit the theory. Yet, the major difference with push-forward models (as GANs or VAEs) is that the optimisation is not directly performed on the push-forward mapping itself but on an auxiliary function (the score). We empirically show that SGMs are able to generate multimodal distributions, which suggests, in the light of our theory, that the structure of the generation dynamic in these models is particularly adequate to (indirectly) learn push-forward mappings with large Lipschitz constants.
***
We modified the paper to make these contributions clearer. More precisely, the following changes have been made in the revised version of the paper:
- In the introduction, we have emphasized that in SGMs, the generated distribution is also a push-forward of a Gaussian, even if these models are not push-forward models in the strict sense of the term, since $ g $ is not a simple neural network anymore. We therefore refer to them as **indirect push-forward models**.
- The paragraph “Contributions of the paper” has been rewritten in order to clarify the  actual contributions of the paper.
***
Other minor changes/additions:
- The bound of Theorem 7 has been rewritten in a more elegant way.
- A corollary of Theorem 7 when $ \nu $ is a mixture of two isotropic Gaussians has been added in the supplementary material.
- A generalization of Corollary 5 when there are more than two disconnected manifold has been added in the supplementary material.
- An additional experiment where we study the influence of the architecture of the generative network on the training stability has been added in the supplementary material.

---

### Comment · Area_Chair_cr6A · 2022-08-10
**Rebuttal Acknowledgement**



Dear Reviewers,

We are entering the discussion phase, where the authors will be not involved in the discussion.

I would like to request you to confirm that you have already read the rebuttal from the authors.

Best

AC

---

> ### Comment · Reviewer_k9qb · 2022-08-10
> **Confirmation**
>
> Dear AC,
>
> I have read the rebuttal of the authors, and I would like to keep my initial score due to the reasons I have pointed out in my discussions with the authors.

---

### Meta-Review · Area_Chair_cr6A · 2022-08-27

**Recommendation:** Accept
**Confidence:** Less certain

**Metareview:**

This paper proposes a theory on the Lipschitz continuity of the pushforward mapping, which transforms a Gaussian distribution to a multimodal distribution. This is a borderline paper. The reviewers are generally positive about the paper and leaning towards acceptance, though there is still some notable gap between the proposed theory and empirical results. The authors are expected to further revise their paper based on the reviewers' suggestions.

**Award:**

No

---

### Decision · Program_Chairs · 2022-09-14

Accept